# Synapse-specific and compartmentalized expression of presynaptic homeostatic potentiation

Xiling Li[1,2], Pragya Goel[1,3], Catherine Chen[1], Varun Angajala, Xun Chen[2], Dion K Dickman*

[1]Department of Neurobiology, University of Southern California, Los Angeles, United States; [2]Neuroscience Graduate Program, University of Southern California, California, United States; [3]Graduate Program in Molecular and Computational Biology, University of Southern California, California, United States

**Abstract** Postsynaptic compartments can be specifically modulated during various forms of synaptic plasticity, but it is unclear whether this precision is shared at presynaptic terminals. Presynaptic homeostatic plasticity (PHP) stabilizes neurotransmission at the *Drosophila* neuromuscular junction, where a retrograde enhancement of presynaptic neurotransmitter release compensates for diminished postsynaptic receptor functionality. To test the specificity of PHP induction and expression, we have developed a genetic manipulation to reduce postsynaptic receptor expression at one of the two muscles innervated by a single motor neuron. We find that PHP can be induced and expressed at a subset of synapses, over both acute and chronic time scales, without influencing transmission at adjacent release sites. Further, homeostatic modulations to CaMKII, vesicle pools, and functional release sites are compartmentalized and do not spread to neighboring pre- or post-synaptic structures. Thus, both PHP induction and expression mechanisms are locally transmitted and restricted to specific synaptic compartments.
DOI: https://doi.org/10.7554/eLife.34338.001

*For correspondence:
dickman@usc.edu

Competing interests: The authors declare that no competing interests exist.

## Introduction

Synaptic strength can be modulated with a remarkable degree of specificity to enable the flexibility necessary for learning and memory, where compartmentalized changes in dendritic spines tune responses to neurotransmitter release during information transfer in the nervous system. Such plasticity mechanisms require compartmentalized trafficking and insertion of glutamate receptors (GluRs) into postsynaptic densities at specific locations in response to correlated activity (*Herring and Nicoll, 2016*; *Malinow, 2003*). However, these processes of Hebbian plasticity are inherently destabilizing, and homeostatic mechanisms have been proposed to adaptively counteract such forces to maintain synaptic strength within physiologically stable levels (*Davis, 2013*; *Pozo and Goda, 2010*; *Turrigiano, 2012*). Although the induction and expression of various forms of plasticity can clearly be restricted to individual postsynaptic compartments, it is less certain that such plasticity can be similarly compartmentalized at the presynaptic terminals of a single neuron. There is evidence for heterogeneity in presynaptic efficacy (*Branco et al., 2008*; *Dobrunz and Stevens, 1997*; *Holderith et al., 2012*; *Trommershäuser et al., 2003*; *Vitureira et al., 2011*), and release probability can vary considerably along a single axon (*Murthy et al., 1997*; *Paul et al., 2015*; *Peled and Isacoff, 2011*). Further, target-specific differences in presynaptic function (*Frank, 1973*; *Katz et al., 1993*; *Scanziani et al., 1998*) and homeostatic plasticity (*Davis and Goodman, 1998*) have been demonstrated. However, how presynaptic terminals are modulated by Hebbian and homeostatic

**eLife digest** Everything we think and do is the result of communication between neurons. This communication takes place at junctions called synapses. When two nerve cells or neurons communicate at a synapse, the output terminal of the first cell releases a chemical called a neurotransmitter. This binds to receiver proteins, or receptors, on the second cell. When this communication is interrupted, synapses can adapt to maintain a stable dialogue between them. This can occur in two ways. Either the first neuron starts to release more neurotransmitter from its output terminal, or the second neuron produces extra receptors with which to detect the neurotransmitter.

But how specific are these changes? The brain contains far more synapses than neurons because each neuron can form synapses with many other cells. Can a neuron adjust how much of the neurotransmitter it releases at some of its synapses while leaving the others unchanged?

Li et al. have now addressed this question by studying a special type of synapse that forms between neurons and muscles, known as a neuromuscular junction. At one particular neuromuscular junction in fruit flies, a single neuron splits into two output terminals, each of which forms a synapse with a different muscle. Li et al. show that when the number of neurotransmitter receptors in one of the muscles is artificially reduced, the associated output terminal compensates by increasing its neurotransmitter release. By contrast, the other output terminal remains unaffected.

This suggests that a neuron can induce remarkably specific changes in a subset of its synapses. This discovery paves the way towards identifying the smallest possible unit of change that can occur in the neurons' ability to communicate. This unit may in turn be the smallest change that can support learning. Such knowledge will help us understand how the nervous system processes and stabilizes information transfer, both in health and after injury or disease.
DOI: https://doi.org/10.7554/eLife.34338.002

forces and whether these adaptations can occur without 'spreading' to adjacent synapses remains enigmatic.

The *Drosophila* neuromuscular junction (NMJ) is a powerful model system to interrogate the mechanisms governing homeostatic synaptic plasticity. At this glutamatergic synapse, genetic and pharmacological perturbations to postsynaptic receptors initiates a retrograde, trans-synaptic signaling system that homeostatically increases presynaptic neurotransmitter release to maintain stable levels of synaptic strength (*Davis and Müller, 2015*; *Frank, 2014*). This form of plasticity is achieved through an increase in presynaptic efficacy and is therefore referred to as presynaptic homeostatic potentiation (PHP). Parallel forms of homeostatic regulation are conserved at NMJs of rodents (*Wang et al., 2016b*) and humans (*Cull-Candy et al., 1980*). PHP initiates a single retrograde signaling system that triggers two key expression mechanisms to enhance presynaptic glutamate release: increases in both presynaptic calcium influx and the number of synaptic vesicles participating in the readily releasable pool (*Goel et al., 2017*; *Kiragasi et al., 2017*; *Müller and Davis, 2012*; *Weyhersmüller et al., 2011*). Several genes necessary for PHP expression have been identified that function as putative retrograde signals (*Orr et al., 2017*; *Wang et al., 2014*) and presynaptic effectors in the motor neuron (*Bruckner et al., 2017*; *Dickman and Davis, 2009*; *Dickman et al., 2012*; *Kiragasi et al., 2017*; *Müller et al., 2015*; *Müller et al., 2012*; *Tsurudome et al., 2010*; *Younger et al., 2013*), but the postsynaptic induction mechanisms that initiate PHP signaling remain enigmatic (*Chen and Dickman, 2017*; *Goel et al., 2017*). Indeed, PHP induction and expression at a subset of synapses within a single motor neuron has never been demonstrated at the *Drosophila* NMJ.

PHP is expressed exclusively at one of two motor neuron subtypes that innervate most muscles at the *Drosophila* NMJ, Type Is and Type Ib. Type Is motor inputs exhibit smaller boutons, less subsynaptic reticulum (SSR), higher basal release probability, and do not participate in PHP adaptation over chronic time scales (*Lnenicka and Keshishian, 2000*; *Newman et al., 2017*). In contrast, Type Ib motor neurons have larger boutons, more elaborate SSR, and lower basal release probability, which is enhanced after loss of postsynaptic GluRs (*Newman et al., 2017*), demonstrating that PHP is expressed exclusively at Type Ib synapses. Further, a reduction in phosphorylated (active) levels of CaMKII, presumably related to PHP inductive signaling, occurs specifically in the postsynaptic

density of Ib boutons (*Goel et al., 2017*; *Newman et al., 2017*), suggesting a possible mechanism for the specificity of retrograde PHP signaling to the Ib motor neuron. Despite these insights, it is not known whether PHP can be expressed at a subset of Type Ib boutons within a single motor terminal, nor whether PHP modulations at individual boutons influence neighboring synapses.

We have developed a genetic manipulation that enables the reduction of postsynaptic GluR expression on one of the two muscles innervated by a single Type Ib motor neuron at the *Drosophila* NMJ. We have used this system to test whether PHP signaling is synapse specific and to determine to what extent the postsynaptic induction and presynaptic expression of PHP is compartmentalized. This analysis has revealed highly specific and compartmentalized PHP adaptations that are restricted and target specific without influencing neurotransmission at neighboring synapses within the same motor neuron.

## Results

### Reduced expression of the glutamate receptor subunit *GluRIIA* specifically on muscle six at the *Drosophila* NMJ

The postsynaptic response to glutamate release at the *Drosophila* NMJ is mediated by two types of GluRs. Both types contain the essential subunits GluRIIC, GluRIID, and GluRIIE, but differ in containing either GluRIIA or GluRIIB subunits (*DiAntonio et al., 1999*; *Featherstone et al., 2005*; *Han et al., 2015*; *Marrus et al., 2004*). Although null mutations in the *GluRIIA* subunit have been studied for decades (*Petersen et al., 1997*), RNAi knock-down of this receptor has not been reported or characterized. We obtained an RNAi transgene targeting the *GluRIIA* subunit (see Mateials and methods) and compared the impact of postsynaptic knock-down of *GluRIIA* by the muscle driver *G14-Gal4* (G14 >GluRIIA^RNAi^) to *GluRIIA* null mutants (*Figure 1*). First, we immunostained the NMJ of wild type, *GluRIIA* mutants, and G14 >GluRIIA^RNAi^ with antibodies against the GluRIIA subunit as well as the common subunits GluRIIC and GluRIID (*Figure 1A*). This revealed an absence of GluRIIA signals from *GluRIIA* mutants, as expected, with signals from GluRIIC and GluR-IID persisting due to the remaining GluRIIB-containing receptors (*Figure 1A,B*). Similarly, GluRIIA expression is almost completely absent in G14 >GluRIIA^RNAi^, with no significant difference in fluorescence intensity compared to *GluRIIA* mutants (*Figure 1A,B*). Indeed, quantitative PCR analysis revealed a dramatic reduction in the level of transcripts encoding the GluRIIA subunit in G14 >GluRIIA^RNAi^, while levels of the other four subunits were not significantly changed (*Figure 1C*). In addition, we quantified synaptic growth in these genotypes, finding a small reduction in bouton number in both *GluRIIA* mutants and G14 >GluRIIA^RNAi^ compared to controls (*Figure 1—figure supplement 1*), as reported previously (*Choi et al., 2014*; *Schmid et al., 2006*; *Sigrist et al., 2002*). Finally, we examined synaptic physiology, which revealed a large reduction in mEPSP amplitude in both *GluRIIA* mutants and G14 >GluRIIA^RNAi^ compared to wild type and *G14-Gal4/+* (*Figure 1D,E* and *Supplementary file 1*), while EPSP amplitudes were not significantly changed between these genotypes because of a homeostatic enhancement in presynaptic glutamate release (quantal content; *Figure 1D,H*). Together, this demonstrates that postsynaptic knock down of the *GluRIIA* subunit effectively phenocopies *GluRIIA* mutants and induces the robust expression of PHP.

Next, we sought to specifically knock down the *GluRIIA* subunit on one of the two muscles innervated by a single Ib motor neuron at the NMJ by selectively biasing expression of *Gal4*. At the muscle 6/7 NMJ, a single Ib and a single Is motor neuron bifurcates to innervate both muscle 6 and 7, with ~60% of the boutons from each motor neuron subtype innervating the larger muscle 6, and ~40% innervating the smaller muscle 7 (*Figure 2—figure supplement 1A,B,C*). To bias *Gal4* expression selectively on muscle 6, we modified a genetic manipulation using the H94-Gal4 driver, which expresses transiently on muscle six early in development (*Davis and Goodman, 1998*). *Gal4* expression on muscle six is amplified and converted into constitutive expression by utilizing a cassette in which a flippase is co-expressed to excise a stop codon between the strong and ubiquitous *Tubulin*-promotor and *Gal4* (see Mateials and methods; [*Choi et al., 2014*; *Roy et al., 2007*]). Thus, this manipulation enables *Gal4* to be strongly and consistently expressed specifically on muscle 6 (*Choi et al., 2014*). We validated this approach by visualizing UAS-GFP selectively on muscle 6, which demonstrated strong expression on muscle six and no detectable expression in either the motor neuron or the adjacent muscle 7 (*Figure 2—figure supplement 1A*). Importantly, we

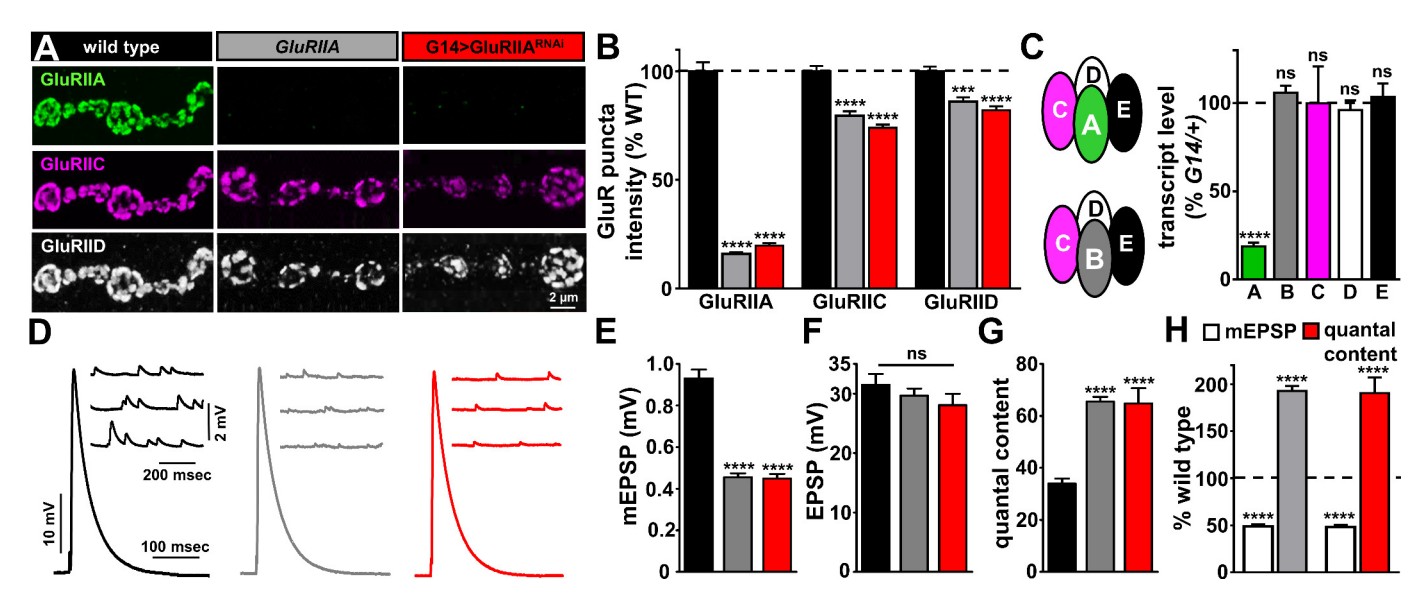

**Figure 1.** *GluRIIA* knock down phenocopies *GluRIIA* mutants. (**A**) Representative images of NMJs on muscle six immunostained with antibodies that recognize GluRIIA, GluRIIC, and GluRIID receptor subunits in wild type (*w1118*), *GluRIIA* mutants (*w1118;GluRIIASP16*), and G14 > GluRIIARNAi (*w1118;G14-Gal4/+;UAS-GluRIIARNAi/+*). (**B**) Quantification of the mean fluorescence intensity of individual GluR puncta reveals GluRIIA subunits are virtually undetectable at NMJs of both *GluRIIA* mutants and G14 > GluRIIARNAi, while the essential subunits GluRIIC and GluRIID are moderately reduced, reflecting expression of the remaining GluRIIB-containing receptors. (**C**) Left: Schematic illustrating the composition of GluRIIA-containing and GluRIIB-containing postsynaptic receptor subtypes at the *Drosophila* NMJ. Right: Quantitative PCR analysis of GluR transcript levels for GluRIIA/B/C/D/E subunits in G14 >GluRIIARNAi normalized to *G14/+*. (**D**) Representative electrophysiological traces of EPSP and mEPSP recordings in the indicated genotypes. (**E–G**) Quantification of mEPSP amplitude (**E**), EPSP amplitude (**F**), and quantal content (**G**) in the indicated genotypes. Note that while mEPSP amplitudes are reduced to similar levels in *GluRIIA* mutants and G14 > GluRIIARNAi, EPSP amplitudes remain similar to wild type because of a homeostatic increase in presynaptic glutamate release (quantal content). (**H**) Quantification of mEPSP amplitude and quantal content values of the indicated genotypes normalized to wild-type values. Error bars indicate ± SEM. Asterisks indicate statistical significance using one-way analysis of variance (ANOVA), followed by Tukey's multiple-comparison test: (****) $p < 0.0001$, (ns) not significant. Detailed statistical information for represented data (mean values, SEM, n, p) is shown in *Supplementary file 1*.

DOI: https://doi.org/10.7554/eLife.34338.003

The following figure supplement is available for figure 1:

**Figure supplement 1.** Synaptic growth is mildly reduced at NMJs of *GluRIIA* mutants and G14 > GluRIIARNAi.

DOI: https://doi.org/10.7554/eLife.34338.004

confirmed that M6 > Gal4 driving a control RNAi (M6 > mCherryRNAi) did not have any significant impact on muscle surface area, synaptic growth, active zone numbers, or synaptic physiology compared to wild type (*Figure 2—figure supplement 1A–H*). Thus, M6 > Gal4 enables strong and biased expression of *Gal4* on muscle six without impacting synaptic growth or function.

Finally, we evaluated this M6 >Gal4 system to determine whether the *GluRIIA* subunit could be specifically knocked down on muscle 6, as observed using pan-muscle knock down in *Figure 1* (see schematic in *Figure 2A*). First, we performed immunocytochemistry at the larval NMJ with antibodies that label the neuronal membrane (HRP) and the GluRIIA subunit in wild type and following knock down of *GluRIIA* on muscle 6 (M6 > GluRIIARNAi). We observed a near-absence of the GluRIIA signal specifically on muscle 6, while GluRIIA expression on the adjacent muscle seven was unperturbed (*Figure 2B,C*). Quantification of synaptic growth on muscles 6 and 7 in M6 > GluRIIARNAi revealed no significant change on muscle 7, while a small but significant reduction was observed on muscle 6, as expected (*Figure 2D*). Finally, quantification of GluRIIA and GluRIID fluorescence levels at muscles 6 and 7 confirmed a large reduction of GluRIIA expression on muscle six and no significant change on muscle seven in M6 > GluRIIARNAi (*Figure 2C,E*). Thus, M6 > GluRIIARNAi effectively and specifically eliminates GluRIIA expression on muscle six without altering glutamate receptor expression on the adjacent muscle 7.

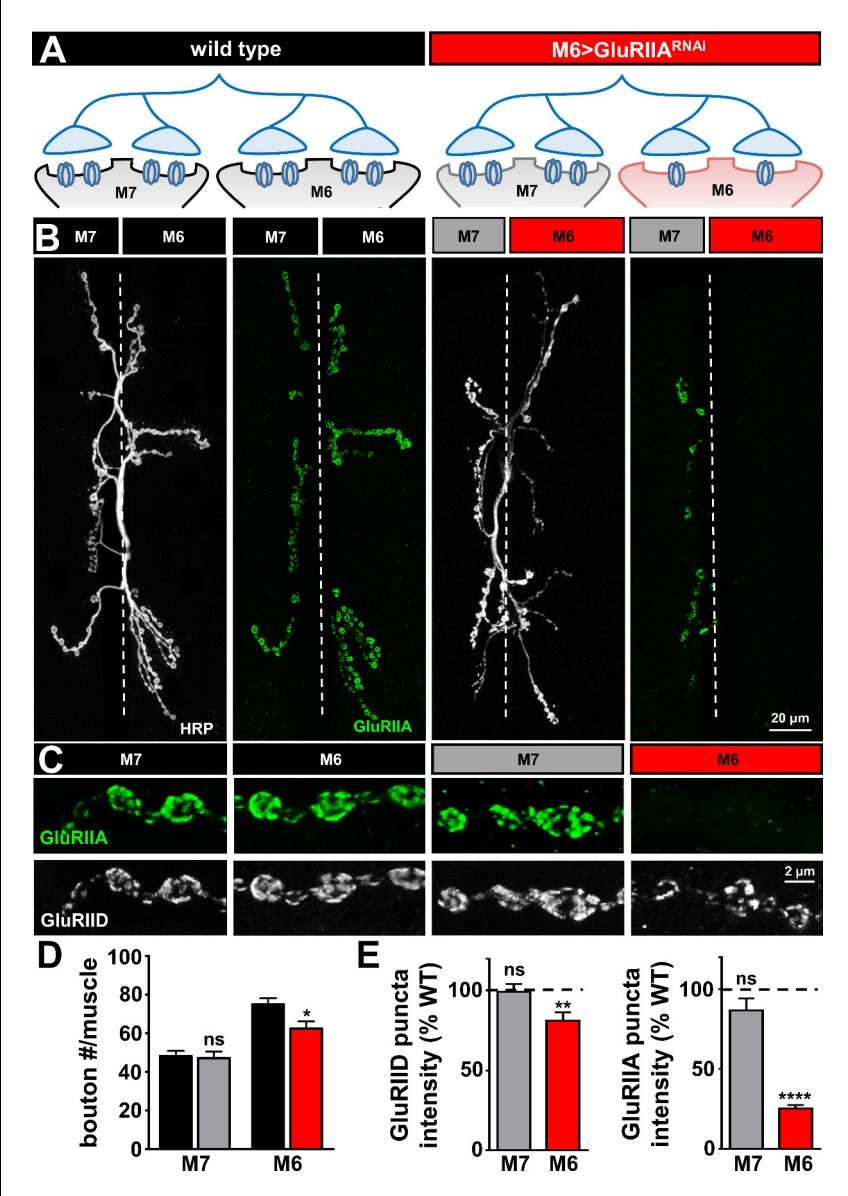

**Figure 2.** GluRIIA-containing receptors can be knocked down specifically on muscle six using M6 >Gal4. (**A**) Schematic of the Type Ib motor neuron innervating both muscles 6 and 7 at the *Drosophila* NMJ. *Gal4* is expressed specifically on muscle six in combination with *UAS-GluRIIA^RNAi* using M6 > Gal4 (M6 > GluRIIA^RNAi: *w^1118;Tub-FRT-STOP-FRT-Gal4, UAS-FLP, UAS-CD8-GFP/+; H94-Gal4, nSyb-Gal80/UAS-GluRIIA^RNAi*). *GluRIIA* expression is reduced on muscle 6, while expression of *GluRIIA* on muscle 7 is unperturbed. (**B**) Representative images of muscle 6/7 NMJs immunostained with antibodies that recognize the neuronal membrane (HRP) and the postsynaptic GluR subunit GluRIIA in wild type and M6 > GluRIIA^RNAi. (**C**) Representative images of individual Ib synaptic boutons immunostained with anti-GluRIIA and anti-GluRIID on the indicated muscles in wild type and M6 > GluRIIA^RNAi. (**D**) Quantification of bouton numbers on muscles 6 and 7 in wild type and M6 > GluRIIA^RNAi. (**E**) Quantification of mean fluorescence intensity of GluRIIA and GluRIID puncta normalized to wild type on the indicated muscles. Asterisks indicate statistical significance using a Student's t test: (*) p<0.05; (**) p<0.01; (****) p<0.0001, (ns) not significant. Error bars indicate ± SEM. Detailed statistical information for represented data (mean values, SEM, (n, p) is shown in *Supplementary file 1*.

DOI: https://doi.org/10.7554/eLife.34338.005

The following figure supplement is available for figure 2:

**Figure supplement 1.** Biased Gal-4 expression on muscle six does not alter synaptic growth or function.
DOI: https://doi.org/10.7554/eLife.34338.006

## Presynaptic homeostatic potentiation can be exclusively expressed at synapses innervating muscle six in M6 > GluRIIA^RNAi

A single Ib motor neuron (RP3) bifurcates to innervate both muscles 6 and 7 at the *Drosophila* NMJ (*Broadie and Bate, 1993*) (*Figure 3A*). Having established strong and selective knock down of the *GluRIIA* subunit on muscle six in M6 > GluRIIA^RNAi, we next characterized synaptic function and homeostatic plasticity. As expected, mEPSP amplitudes on muscle 6 were diminished at M6 > GluRIIA^RNAi NMJs, while mEPSP amplitudes were not affected on the adjacent muscle 7 (*Figure 3A–C*). We considered three possibilities for how presynaptic neurotransmitter release sites may be modulated within the single Ib terminal in response to *GluRIIA* knock down exclusively on muscle 6. First, if PHP signaling is communicated to synapses in the Ib motor neuron innervating muscle 6, but the entire motor neuron undergoes PHP adaptations, then quantal content would be enhanced on both muscles 6 and 7. Second, PHP signaling may be communicated to synapses innervating only muscle 6, but PHP expression may be occluded without simultaneous signaling also received from muscle 7, leading to no change in quantal content on either muscle. Finally, if PHP signaling is target-specific and compartmentalized, then quantal content should be selectively enhanced on synapses innervating muscle 6 in response to reduced *GluRIIA* expression, while synaptic function at synapses innervating the adjacent muscle seven would be unchanged. Results of electrophysiological recordings were consistent with this last model: EPSP amplitude was similar on both muscles 6 and 7 in M6 > GluRIIA^RNAi and not significantly different compared to wild type (*Figure 3A,D*). Indeed, quantal content was selectively enhanced only at synapses innervating muscle 6, while quantal content at synapses innervating muscle seven was unaffected (*Figure 3B,E*).

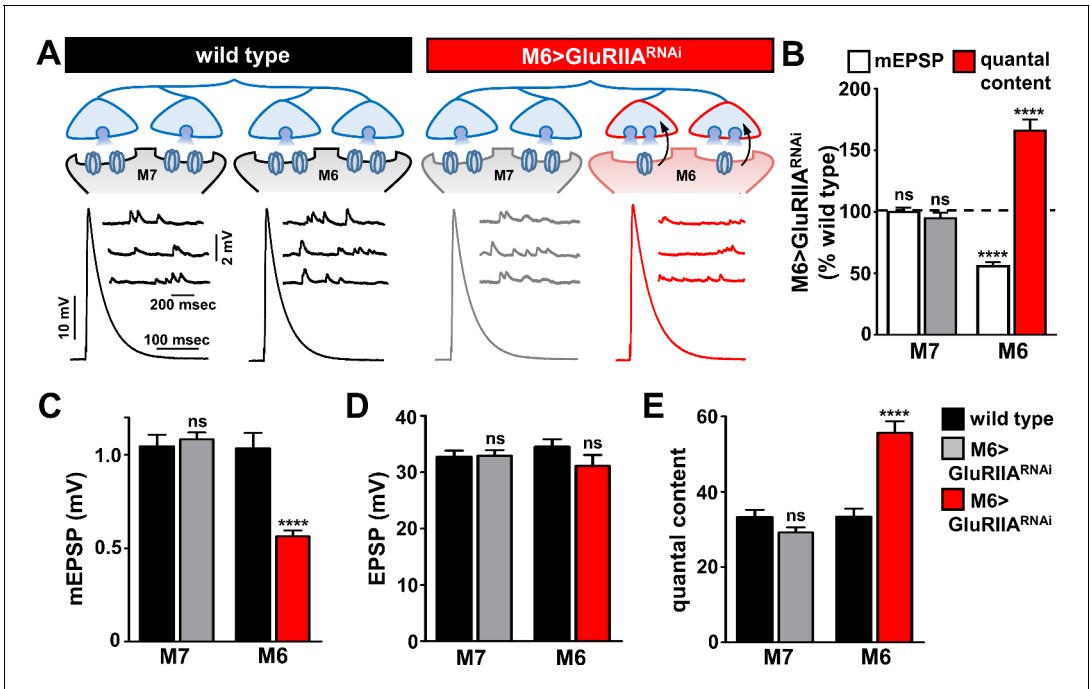

**Figure 3.** Presynaptic homeostatic potentiation can be induced and expressed exclusively at synapses innervating muscle 6. (**A**) Schematic and electrophysiological traces of recordings from muscles 7 and 6 of wild type and M6 > GluRIIA^RNAi NMJs. Note that mEPSPs are reduced only on muscle 6 of M6 > GluRIIA^RNAi, while EPSP amplitudes are similar across all muscles. Thus, the expression of PHP is restricted to synapses innervating muscle 6 of M6 > GluRIIA^RNAi and does not impact neurotransmission at neighboring synapses within the same motor neuron. (**B**) Quantification of mEPSP amplitude and quantal content values of M6 > GluRIIA^RNAi normalized to wild-type values. (**C–E**) Quantification of mEPSP amplitude (**C**), EPSP amplitude (**D**), and quantal content (**E**) in the indicated muscles and genotypes. Asterisks indicate statistical significance using a Student's t test: (****) p<0.0001, (ns) not significant. Error bars indicate ± SEM. Detailed statistical information for represented data (mean values, SEM, (**n, p**) is shown in ***Supplementary file 1***.

DOI: https://doi.org/10.7554/eLife.34338.007

Thus, PHP can be induced and expressed exclusively at a subset of synapses within the same motor neuron without influencing neurotransmitter release at neighboring sites.

The results above suggest that while PHP was chronically induced and expressed specifically on synapses innervating muscle 6, the adjacent synapses within the same motor neuron that innervate muscle seven were apparently not affected. One possibility is that PHP adaptations are induced throughout the entire motor neuron innervating both muscles 6 and 7, but that negative regulators are active that repress or occlude the expression of PHP on synapses innervating muscle 7. Indeed, such a model has been proposed (*Müller et al., 2011*). In this case, PHP should not be capable of being induced or expressed at the synapses innervating muscle seven in M6 > GluRIIA$^{RNAi}$. PHP can be acutely induced and expressed through a pharmacological blockade of the postsynaptic GluRs using a 10-min incubation in the presence of the antagonist philanthotoxin-433 (PhTx; [*Frank et al., 2006*]). This results in an acute reduction in mEPSP amplitude due to blockade of GluRIIA-containing receptors in the postsynaptic muscle, but EPSP amplitudes are maintained because of PHP expression. We reasoned that acute application of PhTx to M6 > GluRIIA$^{RNAi}$ NMJs would enable us to determine whether PHP could be induced and expressed at synapses innervating muscle 7 following chronic expression of PHP on the adjacent synapses innervating muscle 6.

We therefore applied PhTx to wild type and M6 > GluRIIA$^{RNAi}$ synapses. As expected, this caused a large reduction in mEPSP amplitudes at wild-type muscles 6 and 7, as well as muscle 7 in M6 > GluRIIA$^{RNAi}$, while a small reduction in mEPSP amplitude was observed at muscle 6 (*Figure 4A–C*). Interestingly, EPSP amplitudes at both muscles 6 and 7 in wild type and M6 > GluRIIA$^{RNAi}$ NMJs were maintained at similar levels (*Figure 4A,D*) due to the homeostatic enhancement of quantal content (*Figure 4B,E*). Together, this demonstrates that PHP can be acutely induced and chronically expressed at distinct presynaptic release sites within the same neuron according to the state of GluR functionality at postsynaptic compartments opposing these sites.

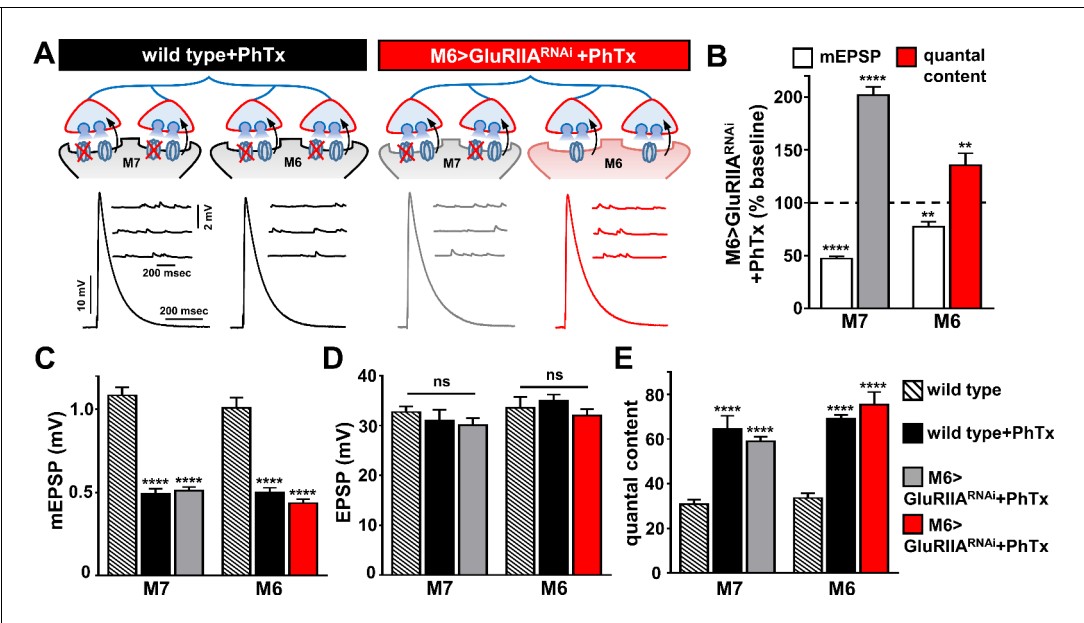

**Figure 4.** PHP can be induced and expressed acutely on muscle 7 following chronic *GluRIIA* knock down on M6. (**A**) Schematic and representative traces illustrating the acute application of PhTx on wild type and M6 >GluRIIA$^{RNAi}$ NMJs. The acute expression of PHP was observed on previously non-potentiated muscle 7 synapses in M6 >GluRIIA$^{RNAi}$ NMJs. mEPSP amplitudes are diminished at wild type and M6 >GluRIIA$^{RNAi}$ NMJs on both muscles following PhTx application, while EPSP amplitudes are maintained at baseline levels due to a homeostatic increase in presynaptic neurotransmitter release. (**B**) Quantification of mEPSP and quantal content values normalized to baseline values (-PhTx) at muscles 6 and 7 NMJs of M6 >GluRIIA$^{RNAi}$ larvae. (**C–E**) Quantification of mEPSP amplitude (**C**), EPSP amplitude (**D**), and quantal content (**E**) values in the indicated genotypes. Asterisks indicate statistical significance using a Student's t test and one-way analysis of variance (ANOVA), followed by Tukey's multiple-comparison test: (**) p<0.01, (****) p<0.001, (ns) not significant. Error bars indicate ± SEM. Detailed statistical information for represented data (mean values, SEM, (**n**, **p**) is shown in *Supplementary file 1*.
DOI: https://doi.org/10.7554/eLife.34338.008

## Homeostatic modulations to the synaptic vesicle pool and functional release sites are compartmentalized at presynaptic terminals

One important presynaptic expression mechanism that enables the enhanced efficacy necessary for PHP expression is an increase in the readily releasable synaptic vesicle pool (RRP; [*Kiragasi et al., 2017*; *Müller et al., 2015*; *Weyhersmüller et al., 2011*]). The RRP is defined as the pool of vesicles that are primed and available for immediate release upon strong synaptic stimulation (*Rosenmund and Stevens, 1996*). Although PHP expression appears to be compartmentalized, it is unknown whether conventional homeostatic modulations to the presynaptic terminal are similarly compartmentalized, or rather whether novel mechanisms are utilized in M6 > GluRIIA$^{RNAi}$. In particular, synaptic vesicles are highly mobile and can rapidly traffic between adjacent boutons at presynaptic terminals (*Darcy et al., 2006*; *Kahms and Klingauf, 2018*), and vesicle pools can span multiple presynaptic terminals (*Staras et al., 2010*). Hence, it is possible that changes in the RRP following PHP expression at a subset of presynaptic terminals may influence vesicle pools at neighboring synapses that do not directly experience local PHP signaling. We therefore measured the size of RRP in wild type and M6 > GluRIIA$^{RNAi}$ synapses separately innervating muscles 6 and 7. To determine RRP size, we performed two electrode voltage clamp (TEVC) measurements using high-frequency stimulation (60 Hz) in elevated external calcium concentrations (3 mM) and measured the cumulative EPSC (*Figure 5A–D*; (*Kiragasi et al., 2017*; *Müller et al., 2015*; *Weyhersmüller et al., 2011*). We observed a ~ 65% increase in the estimated RRP size that was restricted to boutons innervating muscle 6 in M6 > GluRIIA$^{RNAi}$, similar in magnitude to what has been reported for muscle 6 synapses in which PHP is expressed following PhTx application (*Müller et al., 2015*) and by loss of *GluRIIA* (*Kiragasi et al., 2017*; *Weyhersmüller et al., 2011*). However, no significant change in RRP size was observed at synapses innervating the adjacent muscle 7 (*Figure 5A,E*). Thus, the homeostatic modulation of the RRP is restricted to presynaptic terminals that oppose postsynaptic compartments with reduced GluR functionality, and does not 'spread' to influence vesicle pools at adjacent release sites.

Next, we examined whether a change in the number of functional release sites (N) accompanies the compartmentalized expression of PHP. N is defined as the number of functional release sites and is one of the three basic parameters used to describe synaptic transmission, where quantal content (QC) is the product of N, P (release probability), and Q (quantal size). Although there is no major difference in the anatomical number of active zones at NMJs of wild-type and *GluRIIA* mutants (*Frank et al., 2006*; *Goel et al., 2017*; *Penney et al., 2012*; *Schmid et al., 2006*), an increase in the fraction of active zones that participate in release has been reported following the expression of PHP (*Davis and Müller, 2015*; *Newman et al., 2017*; *Penney et al., 2016*). Indeed, the value of N is significantly increased following PhTx application and in *GluRIIA*-mutant synapses (*Müller et al., 2012*; *Weyhersmüller et al., 2011*). We determined the number of functional release sites at NMJs of muscles 7 and 6 in wild type, *GluRIIA* mutants, and M6 >GluRIIA$^{RNAi}$ using a variance-mean plot analysis (*Böhme et al., 2016*; *Clements and Silver, 2000*). We performed TEVC recordings over a range of increasing extracellular calcium concentrations, from 0.5 mM to 6 mM (*Figure 6B,F*). The variance in the amplitude of repeated evoked responses fluctuates across different calcium concentrations in relation to the proportion of total release sites that participate in these responses. At low-calcium concentrations, the number of release sites that participate in the evoked response is low, and the variance is therefore small in this condition. As the extracellular calcium concentration is elevated, the variance then increases with increasing release during repeated evoked responses due to an increase in the number of release sites that participate in synaptic transmission. At very high extracellular calcium concentrations, the variance is then reduced due to saturation of the total number of releasable sites.

We plotted the variance of EPSC responses across increasing calcium conditions against the mean of the EPSC amplitude at each individual calcium conditions from recordings at both muscle 7 and muscle 6 (*Figure 6C,G*). This analysis resulted in a parabolic behavior of variance-mean plot due to the binomial nature of the fluctuation (*Clements and Silver, 2000*; *Weyhersmüller et al., 2011*). The number of functional release sites (N) was determined by fitting a parabola to the variance-mean plot (see Mateials and methods). Based on this result, the value of N increased in *GluRIIA*-mutant NMJs compared to wild type at both muscles 7 and 6 (*Figure 6D,H*), consistent with what was reported following acute PHP expression and indicating that chronic PHP expression requires

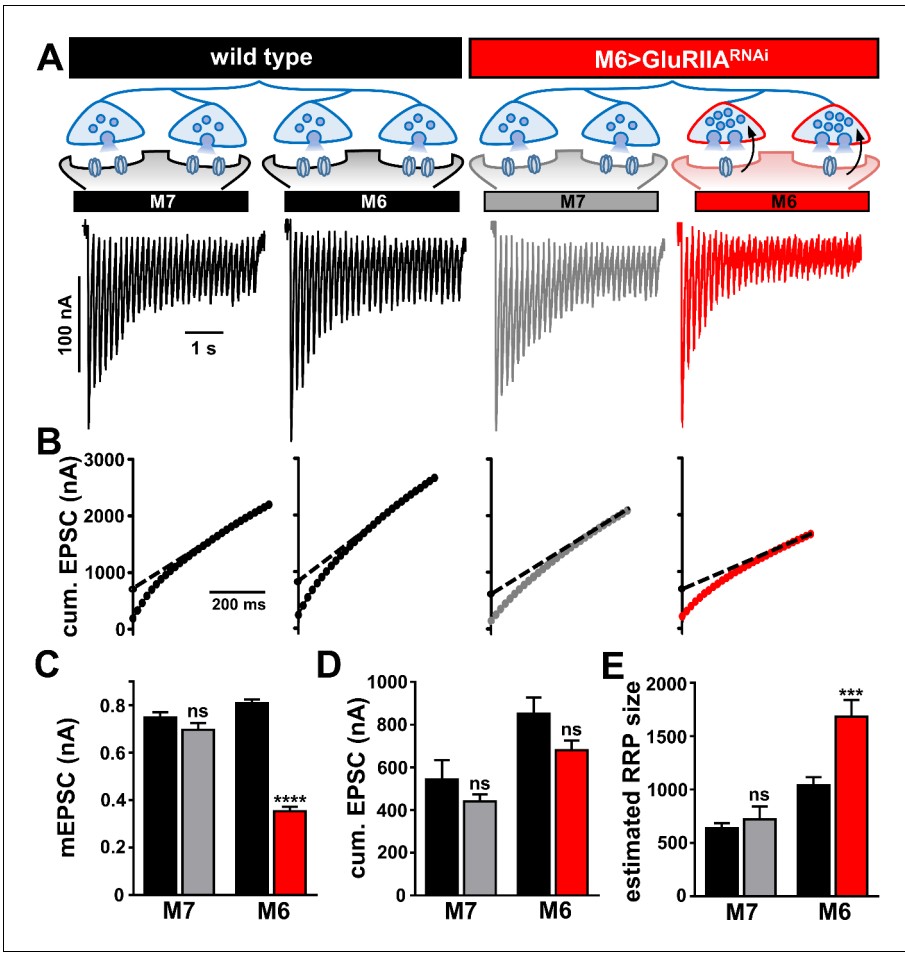

**Figure 5.** Homeostatic modulation of the readily releasable vesicle pool is restricted to synapses innervating muscle 6 at M6 >GluRIIA[RNAi] NMJs. (A) Schematic illustrating the size of the RRP is enhanced specifically at the terminals innervating muscle six in M6 > GluRIIA[RNAi]. Representative traces of two-electrode voltage clamp recordings (30 stimuli at 60 Hz in 3 mM extracellular $Ca^{2+}$) from muscles 7 or 6 in the indicated genotypes. (B) Averaged cumulative EPSC amplitude plotted as a function of time. A line fit to the 18-30[th] stimuli was back-extrapolated to time 0. (C–E) Quantification of mEPSC amplitude (C), average cumulative EPSC values (D), and estimated readily releasable pool (RRP) sizes (E) for the indicated muscles and genotypes. Note that RRP size is significantly increased at muscle 6 NMJs, but no change is observed at muscle 7. Error bars indicate ± SEM. Asterisks indicate statistical significance using a Student's t test: (*) $p<0.05$; (***) $p<0.001$; (ns) not significant. Detailed statistical information for represented data (mean values, SEM, n, p) is shown in **Supplementary file 1**.
DOI: https://doi.org/10.7554/eLife.34338.009

the recruitment of additional functional release sites to participate in presynaptic neurotransmitter release. Further, a similar increase in N was observed at muscle 6 in M6 > GluRIIA[RNAi] (*Figure 6H*), while no significant change was found at muscle 7 (*Figure 6D*), suggesting that the biased induction of PHP results in the compartmentalized expression of the same mechanisms observed in *GluRIIA* mutants (*Figure 6A,E*). Therefore, retrograde PHP expression is achieved by elevating the RRP and recruiting additional functional release sites to participate in transmission, with specificity according to the excitability state of their postsynaptic partners.

## Compartmentalized reductions in postsynaptic CaMKII activity are required for PHP induction in M6 > GluRIIA[RNAi]

The postsynaptic induction mechanisms that drive PHP retrograde signaling are unclear. However, reductions in postsynaptic CaMKII activity have been proposed to mediate the induction of retrograde PHP signaling (*Haghighi et al., 2003*). Indeed, modulations to postsynaptic CaMKII

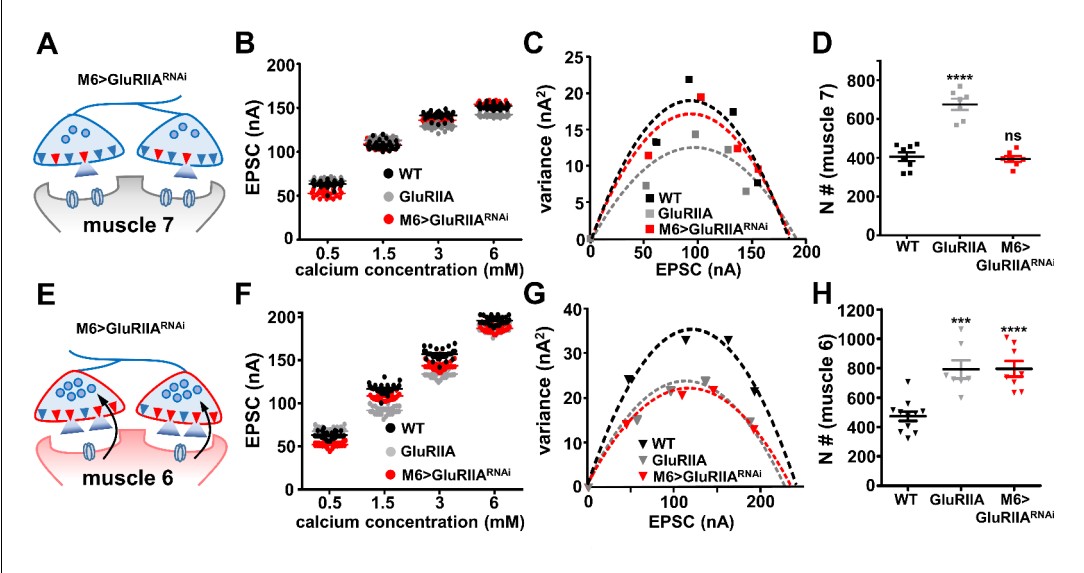

**Figure 6.** The number of functional release sites are specifically enhanced at synapses innervating muscle 6 at M6 >GluRIIA[RNAi] NMJs. (**A**) Schematic illustrating the number of functional release sites (marked as red triangles) at synapses innervating muscle 7 of M6 > GluRIIA[RNAi] NMJs. (**B**) Scatter plot EPSC distribution of recordings derived from muscle 7 of wild type, *GluRIIA* mutants, and M6 > GluRIIA[RNAi] in the indicated extracellular $Ca^{2+}$ concentrations. (**C**) Variance-mean plots for the indicated genotypes. Variance was plotted against the mean amplitude of 30 EPSCs recorded at 0.2 Hz from the four $Ca^{2+}$ concentrations detailed in (**B**). Dashed lines are the best fit parabolas to the data points. (**D**) Estimated number of functional release sites (N) obtained from the variance-mean plots in (**C**). Note that the number of functional release sites are enhanced only at muscle 7 NMJs of *GluRIIA* mutants. (**E**) Schematic illustrating the number of functional release sites (marked as red triangles) at synapses innervating muscle 6 of M6 > GluRIIA[RNAi] NMJs. (**F**) Scatter plot EPSC distribution of recordings derived from muscle 6 of wild type, *GluRIIA* mutants, and M6 > GluRIIA[RNAi] in the indicated extracellular $Ca^{2+}$ concentrations. (**G**) Variance-mean plots for the indicated genotypes. (**H**) Estimated number of functional release sites (N) obtained from the variance-mean plots in (**G**). Both *GluRIIA* mutants and M6 > GluRIIA[RNAi] showed a significant enhancement of functional release site number at muscle 6 NMJs compared to wild type. Error bars indicate ± SEM. Asterisks indicate statistical significance using one-way analysis of variance (ANOVA), followed by Tukey's multiple-comparison test: (***) p<0.001; (****) p<0.0001; (ns) not significant. Detailed statistical information for represented data (mean values, SEM, (n, p) is shown in *Supplementary file 1*.

DOI: https://doi.org/10.7554/eLife.34338.010

phosphorylation has been demonstrated to occur in a synapse-specific and activity-dependent manner at the *Drosophila* NMJ (*Hodge et al., 2006*). Consistent with this idea, reductions in the level of phosphorylated (active) CaMKII were observed specifically at postsynaptic densities of Type Ib boutons following PhTx application and in *GluRIIA* mutants (*Goel et al., 2017*; *Newman et al., 2017*). To determine whether this same signaling system mediates PHP induction and is restricted to specific targets at M6 > GluRIIA[RNAi] NMJs, we examined pCaMKII levels at postsynaptic densities of Is and Ib boutons on both muscles 6 and 7. Ib and Is boutons were distinguished by differential areas and intensity signals of the postsynaptic scaffold Discs large (Dlg). We observed a significant reduction in the mean intensity of pCaMKII in Ib postsynaptic densities only on muscle 6 of M6 > GluRIIA[RNAi], while no significant difference was observed at Ib synapses in the adjacent muscle 7, nor were any changes found at Is synapses on either muscle (*Figure 7A–E*). Thus, postsynaptic pCaMKII levels are diminished and compartmentalized at Ib boutons specifically on muscle 6 without impacting pCaMKII levels on neighboring Is boutons or at the adjacent muscle 7 in M6 > GluRIIA[RNAi].

Finally, we tested whether a reduction in postsynaptic CaMKII activity was required for retrograde PHP signaling. In particular, if PHP signaling is induced at Ib postsynaptic densities through diminished pCaMKII levels, as suggested by immunostaining, then postsynaptic overexpression of a constitutively active, phospho-mimetic form of CaMKII, CaMKII[T287D], should inhibit or occlude PHP induction and expression (*Haghighi et al., 2003*). We first expressed CaMKII[T287D] alone on muscle 6 (M6>CaMKII[T287D]) to determine if baseline synaptic function was influenced by constitutively active CaMKII (*Figure 8A*). We found no significant difference in synaptic physiology on muscles 6 or 7 in this condition (*Figure 8A–D*). Next, we expressed constitutively active CaMKII in combination with

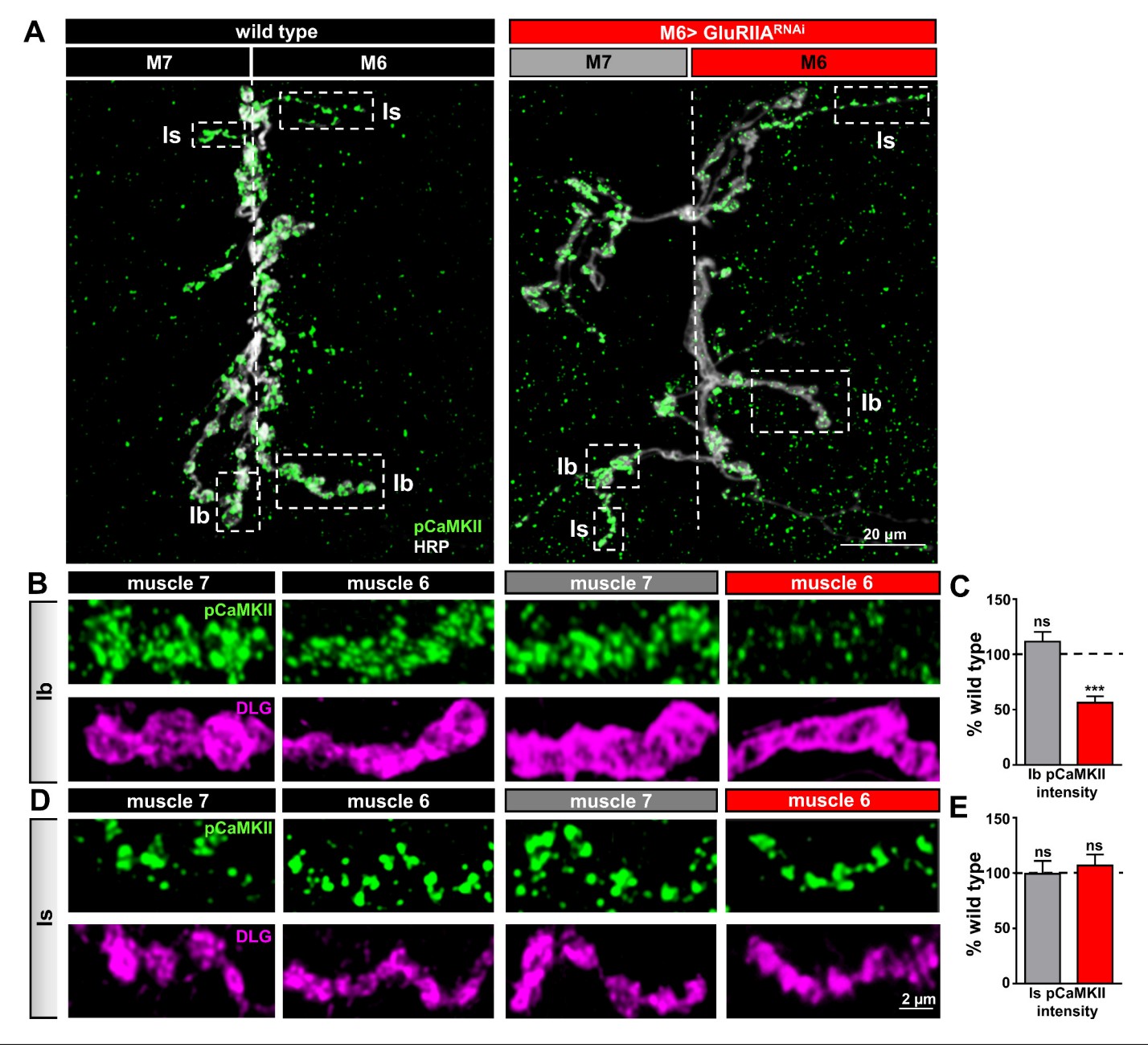

**Figure 7.** Compartmentalized changes in CaMKII activity at postsynaptic densities of M6 > GluRIIA[RNAi] NMJs. (**A**) Representative images of muscle 6/7 NMJs from wild type and M6 > GluRIIA[RNAi] immunostained with antibodies against with the active (phosphorylated) form of CaMKII (pCaMKII) and the presynaptic neuronal membrane marker HRP. Note that pCaMKII intensity is specifically reduced at Ib postsynaptic densities on muscle 6 of M6 >GluRIIA[RNAi] (dashed areas). (**B,C**) Images (**B**) and quantification (**C**) of pCaMKII fluorescence intensity at Ib boutons on muscles 7 and 6 normalized to wild-type values. A reduction in pCaMKII intensity is observed on muscle 6 of M6 > GluRIIA[RNAi], while no significant change is observed at synapses innervating the adjacent muscle 7. (**D,E**) Images (**D**) and quantification (**E**) of pCaMKII intensity at Is boutons shows no significant difference between wild type and M6 > GluRIIA[RNAi] at NMJs of either muscle. Asterisks indicate statistical significance using a Student's t test: (***) p<0.001, (ns) not significant. Error bars indicate ± SEM. Detailed statistical information for represented data (mean values, SEM, (**n, p**) is shown in *Supplementary file 1*.
DOI: https://doi.org/10.7554/eLife.34338.011

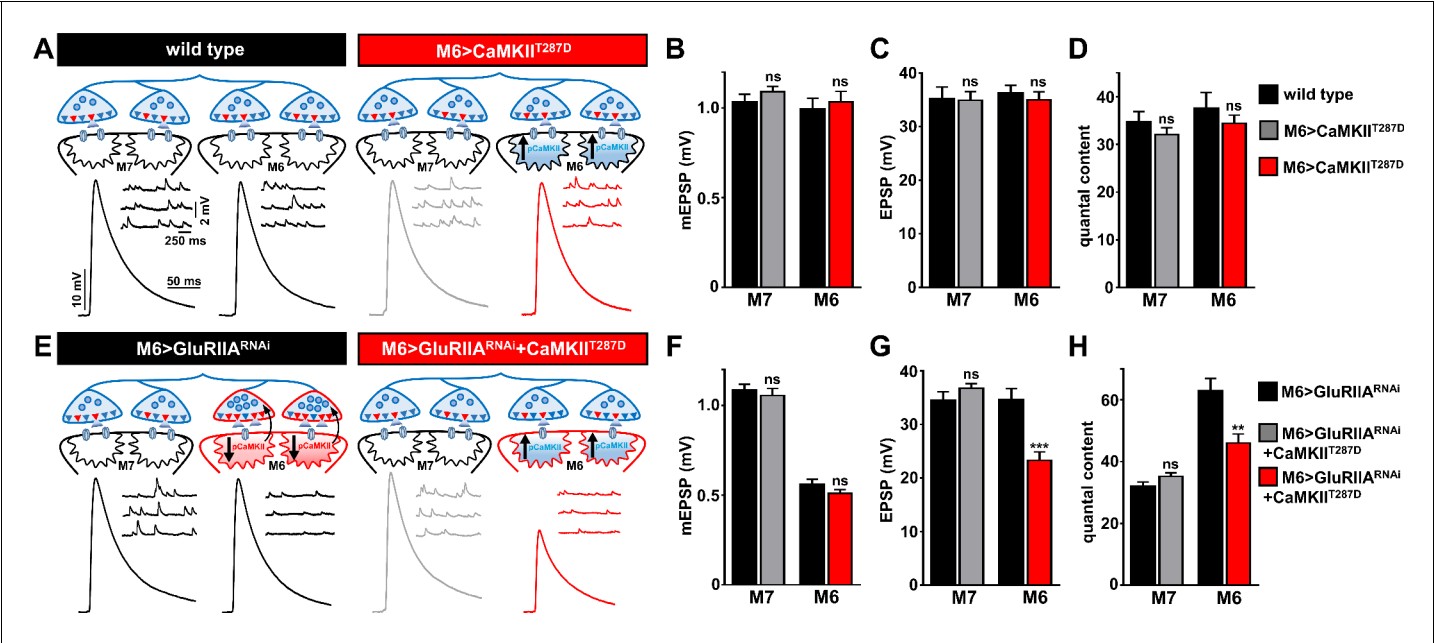

**Figure 8.** Overexpression of constitutively active CaMKII inhibits retrograde PHP signaling at M6 > GluRIIA$^{RNAi}$ NMJs. (A) Schematic and representative traces illustrating that overexpression of a constitutively active form of CaMKII only on muscle 6 (M6>CaMKII$^{T287D}$) has no significant impact on neurotransmission. (B–D) Quantification of mEPSP amplitude (B), EPSP amplitude (C), and quantal content (D) in the indicated muscles. (E) Schematic and representative traces illustrating that expression of constitutively active CaMKII with *GluRIIA* knock down (M6 > GluRIIA$^{RNAi}$ + CaMKII$^{T287D}$) on muscle 6 inhibits the retrograde potentiation of presynaptic glutamate release. (F–H) Quantification of mEPSP amplitude (F), EPSP amplitude (G), and quantal content (H) in the indicated muscles. Asterisks indicate statistical significance using a Student's t test: (**) $p<0.01$; (***) $p<0.001$; (ns) not significant. Error bars indicate ± SEM. Detailed statistical information for represented data (mean values, SEM, n, p) is shown in *Supplementary file 1*.
DOI: https://doi.org/10.7554/eLife.34338.012

*GluRIIA* knock down on muscle 6 (M6 > GluRIIA$^{RNAi}$ + CaMKII$^{T287D}$). If reduced CaMKII activity were functionally required for retrograde PHP signaling, and not just a compartmentalized biomarker of GluRIIA levels and/or activity at Ib boutons, then constitutively active CaMKII should disrupt PHP expression at M6 >GluRIIA$^{RNAi}$ synapses. Indeed, EPSP amplitude was not maintained at baseline levels due to a failure to homeostatically increase quantal content in M6 > GluRIIA$^{RNAi}$ + CaMKII$^{T287D}$ (*Figure 8E–H*). In this condition, quantal content was significantly increased (134 ± 8% increase for M6 > GluRIIA$^{RNAi}$ + CaMKII$^{T287D}$ compared to M6>CaMKII$^{T287D}$) but was below the level necessary to maintain synaptic strength (166 ± 9% increase in quantal content for M6 > GluRIIA$^{RNAi}$ compared to wild type). This data is consistent with reduced CaMKII activity, compartmentalized at Ib postsynaptic densities, being required for retrograde PHP signaling. We propose a model schematized in *Figure 8E*.

## Discussion

Although the genes and mechanisms that mediate retrograde homeostatic potentiation have been intensively investigated, whether this process can be expressed and restricted to a subset of synapses within a single neuron has not been determined. We have developed a manipulation that enables the loss of GluRs on only one of the two postsynaptic targets innervated by a Type Ib motor neuron at the *Drosophila* NMJ. Our analysis of synaptic structure and function in this condition has revealed the spectacular degree of compartmentalization in postsynaptic signaling and presynaptic expression that ultimately orchestrate the synapse-specific modulation of presynaptic efficacy.

### Compartmentalization of postsynaptic PHP signaling

GluRs are dynamically trafficked in postsynaptic compartments where they mediate the synapse-specific expression of Hebbian plasticity such as LTP (*Herring and Nicoll, 2016*; *Matsuzaki et al., 2004*)

and homeostatic plasticity, including receptor scaling (*Hou et al., 2008*; *Pozo and Goda, 2010*; *Sutton et al., 2006*). In contrast, homeostatic plasticity at the human, mouse, and fly NMJ is expressed through a presynaptic enhancement in neurotransmitter release, but is induced through a diminishment of postsynaptic neurotransmitter receptor functionality (*Cull-Candy et al., 1980*; *Frank et al., 2006*; *Petersen et al., 1997*; *Wang et al., 2016b*). Using biased expression of *Gal4* to reduce GluR levels on only one of the two muscle targets innervated by a single motor neuron, we demonstrate that the inductive signaling underlying PHP is compartmentalized at the postsynaptic density, and does not influence activity at synapses innervating the adjacent muscle. Postsynaptic changes in CaMKII function and activity have been associated with PHP retrograde signaling (*Goel et al., 2017*; *Haghighi et al., 2003*; *Newman et al., 2017*). Consistent with this compartmentalized inductive signaling, we observed pCaMKII levels to be specifically reduced at postsynaptic densities of Ib boutons in which GluR expression is perturbed, while pCaMKII was unchanged at postsynaptic compartments opposite to Is boutons and at NMJs in the adjacent muscle with normal GluR expression. Further, postsynaptic overexpression of the constitutively active CaMKII occludes the expression of PHP. Similar synapse-specific control of postsynaptic CaMKII phosphorylation, modulated by activity, has been previously observed (*Hodge et al., 2006*). As noted in other studies (*Goel et al., 2017*; *Newman et al., 2017*), this localized reduction in pCaMKII provides a plausible mechanism for the inductive PHP signaling restricted to and compartmentalized at Ib synapses.

How does a perturbation to GluR function lead to a reduction in CaMKII activity that is restricted to postsynaptic densities opposing Type Ib boutons? Recent evidence suggests that distinct mechanisms regulate pCaMKII levels during retrograde PHP signaling depending on pharmacologic or genetic perturbation to glutamate receptors and the role of protein synthesis (*Goel et al., 2017*). Scaffolds at postsynaptic densities are associated in complexes with GluRs and CaMKII (*Gillespie and Hodge, 2013*; *Hodge et al., 2006*; *Koh et al., 1999*; *Lu et al., 2003*; *Mullasseril et al., 2007*). Intriguingly, the scaffold dCASK is capable of modulating CaMKII activity at specific densities in an activity-dependent fashion (*Hodge et al., 2006*; *Malik et al., 2013*). Further, CaMKII activity can regulate plasticity with specificity at subsets of synapses in *Drosophila* and other systems (*Griffith, 2004*; *Hodge et al., 2006*; *Merrill et al., 2005*). Although we cannot rule out intra-cellular cross talk between Is and Ib boutons, as GluRIIA is reduced at postsynaptic sites of both neuronal subtypes, it is striking that reductions in pCaMKII are restricted to Ib postsynaptic compartments. An attractive model, therefore, is that the postsynaptic density isolates calcium signaling over chronic time scales to compartmentalize PHP induction. The membranous complexity and geometry of the SSR at the *Drosophila* NMJ may be the key to restricting calcium signaling at these sites, as this structure can have major impacts on ionic signaling during synaptic transmission (*Nguyen and Stewart, 2016*; *Teodoro et al., 2013*). These properties, in turn, may lead to local modulation of CaMKII function (*Goel et al., 2017*; *Griffith, 2004*; *Haghighi et al., 2003*; *Newman et al., 2017*). Interestingly, *Drosophila* mutants with defective SSR elaboration and complexity have been associated with defects in PHP expression (*Koles et al., 2015*). In the mammalian central nervous system, it is well established that dendritic spines function as biochemical compartments that isolate calcium signaling while enabling propagation of voltage changes (*Svoboda et al., 1997*; *Yuste and Denk, 1995*), and it is tempting to speculate that the SSR may subserve similar functions at the *Drosophila* NMJ to enable synapse-specific retrograde signaling.

## Compartmentalization of presynaptic PHP expression

The homeostatic modulation of presynaptic neurotransmitter release is compartmentalized at the terminals of Type Ib motor neurons. It was previously known that PHP can be acutely induced and expressed without any information from the cell body of motor neurons (*Frank et al., 2006*). Our data suggests that the signaling necessary for PHP expression is even further restricted to specific postsynaptic densities and presynaptic boutons, demonstrated through several lines of evidence. First, quantal content is specifically enhanced at boutons innervating muscle 6 in M6 > GluRIIA[RNAi] without measurably impacting transmission on the neighboring boutons innervating muscle 7. In addition, PHP can be acutely induced at synapses innervating muscle 7 despite PHP having been chronically expressed at muscle 6. Finally, the homeostatic modulation of the RRP and enhancement of the functional number of release sites is fully expressed regardless of whether PHP is induced at all Type Ib boutons or only a subset. Thus, PHP signaling is orchestrated at specific boutons

according to the state of GluR functionality of their synaptic partners and does not influence neighboring boutons within the same motor neuron.

Although the compartmentalized expression of PHP was not unexpected, there was precedent to suspect inter-bouton crosstalk during homeostatic signaling. In the dynamic propagation of action potentials along the axon, the waveform could, in principle, change following PHP expression to globally modulate neurotransmission at all release sites in the same neuron. However, voltage imaging did not identify any change in the action potential waveform at individual boutons following PHP signaling (*Ford and Davis, 2014*; *Gaviño et al., 2015*), and we did not observe any impact on neighboring boutons despite PHP being induced at a subset of synapses in the same motor neuron. Further, mobilization of an enhanced readily releasable synaptic vesicle pool is necessary for the expression of PHP (*Davis and Müller, 2015*; *Kiragasi et al., 2017*; *Müller et al., 2015*; *Weyhersmüller et al., 2011*), and synaptic vesicles and pools are highly mobile within and between presynaptic compartments (*Darcy et al., 2006*; *Kahms and Klingauf, 2018*; *Staras et al., 2010*). Hence, it was conceivable that a mobilized RRP, induced at some presynaptic compartments, may be promiscuously shared between other boutons. However, while we observed a large enhancement in the RRP at synapses innervating muscle 6 in M6 > GluRIIA$^{RNAi}$, this adaptation had no impact on the RRP at adjacent presynaptic compartments innervating muscle 7. Thus, PHP signaling is constrained to boutons innervating one of two postsynaptic targets and does not 'spread' to synapses innervating the adjacent target despite sharing common cytosol, voltage, and synaptic vesicles.

What molecular mechanisms mediate the remarkable specificity of PHP expression at presynaptic compartments? One attractive possibility is that active zones themselves are fundamental units and act as substrates for the homeostatic modulation of presynaptic function. The active zone scaffold BRP remodels during both acute and chronic PHP expression (*Goel et al., 2017*; *Weyhersmüller et al., 2011*), and other active zone proteins are likely to participate in this remodeling (*Gratz et al., 2018*). Indeed, many genes encoding active zone components are required for PHP expression, including the calcium channel *cac* (*Frank et al., 2006*) and auxiliary subunit $\alpha2-\delta$ (*Wang et al., 2016a*), the *piccolo* homolog *fife* (*Bruckner et al., 2017*), the scaffolds *RIM* (*Rab3-interacting Molecule*; [*Müller et al., 2012*]) and *RIM-binding protein* (*RBP*; [*Müller et al., 2015*]), and the kainite receptor *DKaiR1D* (*Kiragasi et al., 2017*). If individual active zones can undergo the adaptations necessary and sufficient for PHP expression, this would imply that PHP can be induced and expressed with specificity at individual active zones. Indeed, the BRP cytomatrix stabilizes calcium channel levels at the active zone (*Kittel et al., 2006*), and also controls the size of the RRP (*Matkovic et al., 2013*), two key presynaptic expression mechanisms that drive PHP. Further, we and others have observed the recruitment of new functional release sites following both chronic and acute PHP expression (*Davis and Müller, 2015*; *Newman et al., 2017*; *Weyhersmüller et al., 2011*), suggesting that previously silent active zones become 'awakened' and utilized to potentiate presynaptic neurotransmitter release. Interestingly, presynaptic GluRs, localized near active zones, are necessary for PHP expression and have the capacity to modulate release with specificity at individual active zones (*Kiragasi et al., 2017*). Thus, active zones can remodel with both the specificity and precision necessary and sufficient for compartmentalized PHP expression.

If each active zone operates as an independent homeostat to adjust release efficacy in response to target-specific changes, how is information transfer at individual sites integrated to ensure stable and stereotypic global levels of neurotransmission? One speculative possibility is that active zones at terminals of each neuron are endowed with a total abundance of material that is tightly controlled and sets stable global levels of presynaptic neurotransmitter release. Such active zone material may be sculpted with considerable heterogeneity within presynaptic terminals, varying in number, size, and density. Consistent with such a possibility, mutations in the synaptic vesicle component Rab3 exhibit extreme changes in active zone size, number, and density, but stable global levels of neurotransmission (*Graf et al., 2009*). Within this global context, plasticity mechanisms may operate at individual active zones, superimposed as independent homeostats to adaptively modulate synaptic strength. In addition, there is intriguing evidence for the existence of 'nanocolumns' between presynaptic active zones and postsynaptic GluRs that form structural and functional signaling complexes (*Biederer et al., 2017*; *Tang et al., 2016*). One particularly appealing possibility, therefore, is that a dialogue traversing synaptic nanocolumns functions to convey the retrograde signaling and active zone remodeling necessary for PHP expression at individual release sites. Studies in mammalian neurons have revealed parallel links between the functional plasticity of active zones, including their

structure and size, and the homeostatic modulation of neurotransmitter release (*Glebov et al., 2017*; *Matz et al., 2010*; *Murthy et al., 2001*; *Schikorski and Stevens, 2001*). Such intercellular signaling systems are likely to modify synaptic structure and function to not only establish precise pre- and post-synaptic apposition during development, but also to maintain the plasticity necessary for synapses to persist with the flexibility and stability to last a lifetime.

## Materials and methods

### Fly stocks and genetics

*Drosophila* stocks were raised at 25°C on standard molasses food. The $w^{1118}$ strain is used as the wild-type control unless otherwise noted, as this is the genetic background of the transgenic lines and other genotypes used in this study. The following fly stocks were used: *G14-Gal4* (*Aberle et al., 2002*); *GluRIIA$^{RNAi}$* (p{TRiP.JF02647}attP2}; Bloomington Drosophila Stock Center (BDSC)); *UAS-CaMKII$^{T287D}$* (BDSC); *mCherry$^{RNAi}$* (p{VALIUM20-mCherry}attP2}; BDSC); *GluRIIA$^{SP16}$* (*Petersen et al., 1997*); *M6-Gal4* (*Tub-FRT-STOP-FRT-Gal4, UAS-FLP, UAS-CD8-GFP; H94-Gal4, nSyb-Gal80*) (*Choi et al., 2014*).

### Immunocytochemistry

Third-instar larvae were dissected in ice cold 0 $Ca^{2+}$ modified HL-3 saline and immunostained as described (*Kikuma et al., 2017*). Briefly, larvae were fixed in Bouin's fixative (Sigma, St. Louis, MO; HT10132-1L) for 2 min and washed with PBS containing 0.1% Triton X-100 (PBST) for 30 min, then blocked for 1 hr in 5% Normal Donkey Serum (NDS). Following overnight incubation in primary antibodies at 4°C, preparations were washed in PBST, incubated in secondary antibodies for 2 hr, washed and mounted in VectaShield (Vector Laboratories). The following antibodies were used: guinea pig anti-vGlut (1:2000; (*Chen et al., 2017*); rabbit anti-DLG (1:5000; [*Pielage et al., 2005*]); mouse anti-GluRIIA (8B4D2; 1:100; Developmental Studies Hybridoma Bank (DSHB)); rabbit anti-GluRIIC (1:1000; [*Marrus et al., 2004*]); guinea pig anti-GluRIID (1:1000; [*Perry et al., 2017*]); mouse anti-pCaMKII (1:100; MA1-047; Invitrogen); mouse anti-GFP (8H11; 1:100; DSHB); Tetramethylrhodamine (TRITC)-conjugated phalloidin (R415; Thermo Fisher); Alexa Fluor 647-conjugated goat anti-HRP (1:200; Jackson ImmunoResearch). Donkey anti-mouse, anti-guinea pig, and anti-rabbit Alexa Fluor 488-, DyLight 405-, and Cyanine 3 (Cy3)-conjugated secondary antibodies (Jackson ImmunoResearch) were used at 1:400.

### Confocal imaging and analysis

Samples were imaged using a Nikon A1R Resonant Scanning Confocal microscope equipped with NIS Elements software and a 100x APO 1.4NA oil immersion objective using separate channels with four laser lines (405 nm, 488 nm, 561 nm, and 637 nm) as described (*Chen et al., 2017*). All genotypes were immunostained in the same tube with identical reagents, then mounted and imaged in the same session. z-stacks were obtained using identical settings for all genotypes with z-axis spacing between 0.15 μm to 0.2 μm within an experiment and optimized for detection without saturation of the signal. Both Type Ib and Is boutons were counted using vGlut and HRP-stained NMJ terminals on muscle 6/7 of segment A3, considering each vGlut puncta to be a bouton. Muscle surface area was calculated by creating a mask around the phalloidin channel that labels the entire muscle. The general analysis toolkit in the NIS Elements software was used to quantify GluRIIA/C/D and pCamKII intensity levels by applying intensity thresholds and filters to binary layers for each the channels of the maximum intensity projection images. To quantify GluR intensity levels, the total fluorescence intensity of each GluRIIA, GluRIIC, or GluRIID puncta was averaged over the NMJ area covered by muscle 6 or muscle seven separately to determine mean fluorescence intensity; this value was then normalized as a percentage of wild type for the corresponding muscle. For analysis of pCaMKII levels, Ib and Is regions were identified using DLG and HRP on muscle 6/7 of segment A2 and A3, and only the pCamKII signal that co-localized with DLG was summated and divided by the bouton area under consideration to obtain average pCamKII. *quantitative PCR:* quantitative PCR (qPCR) was performed using the Luna Universal One-Step RT-qPCR Kit (NEB, E3005S) according to the manufacturer's instructions. RNA was isolated and prepared from body wall tissue as described previously (*Chen and Dickman, 2017*). 20 ng of total RNA was used as the template for each reaction. Three

biological replicates were performed for each sample and the comparative Ct method was used for qPCR data analysis. The following primers were used (fwd; rev: 5′−3′):

GluRIIA: TCCTCAACTTGGAACTGGAAAG; CGTACTTTTCCCTGCCTCTG.
GluRIIB: GCGAATACAGATGAATGGGATG; TGCATGAAGGGTACAGTGAAG.
GluRIIC: CGGAAAACTGGACAAGGAAAC; AGCTGCATAAAGGGCACTG.
GluRIID: CCCAAGCTGTCAACTTCAATG; CCATAACCCTGGAACTGATTGT.
GluRIIE: CGGTGCAAAGAAAACTGGATC; GTCTTAACTCGATTCACTCCCTC.
$\alpha$Tub84D (control): CTACAACTCCATCCTAACCACG; CAGGTTAGTGTAAGTGGGTCG.

## Electrophysiology

All dissections and recordings were performed in modified HL-3 saline (*Dickman et al., 2005*; *Kiragasi et al., 2017*; *Stewart et al., 1994*) at room temperature containing (in mM): 70 NaCl, 5 KCl, 10 MgCl$_2$, 10 NaHCO$_3$, 115 Sucrose, 5 Trehelose, 5 HEPES, and 0.4 CaCl$_2$ (unless otherwise specified), pH 7.2. Neuromuscular junction sharp electrode recordings were performed on muscles 6 or 7 of abdominal segments A2 or A3 in wandering third-instar larvae. Biased Gal4 expression was verified by verifying GFP fluorescence on the particular muscle before experimentation, and recordings were performed at only the GFP-positive muscle 6 and the adjacent muscle 7. Muscle input resistance (R$_{in}$) and resting membrane potential (V$_{rest}$) were monitored during each experiment (*Supplementary file 1*). To acutely block postsynaptic receptors, larvae were incubated with or without philanthotoxin-433 (20 μM; Sigma) and resuspended in HL-3 for 10 mins, as described (*Dickman and Davis, 2009*; *Frank et al., 2006*).

The readily releasable pool (RRP) size was estimated by analyzing cumulative EPSC amplitudes while recording using a two-electrode voltage clamp (TEVC) configuration. Muscles were clamped to −70 mV and EPSCs were evoked with a 60 Hz, 60 stimulus train while recording in HL-3 supplemented with 3 mM Ca$^{2+}$. A line fit to the linear phase (stimuli # 18–30) of the cumulative EPSC data was back-extrapolated to time 0. The RRP value was estimated by determining the extrapolated EPSC value at time 0 and dividing this value by the average mEPSC amplitude. More details of the RRP size analysis can be found at Bio-protocol (*Goel et al., 2019*). Data used in the variance-mean plot was obtained from TEVC recordings using an initial 0.5 mM Ca$^{2+}$ concentration, which was subsequently increased to 1.5, 3.0, and 6.0 mM through saline exchange using a peristaltic pump (Langer Instruments, BT100-2J). EPSC amplitudes were monitored during the exchange, and 30 EPSC (0.5 Hz stimulation rate) recordings were performed in each calcium condition. To obtain the variance-mean plot, the variance (squared standard deviation) and mean (averaged evoked amplitude) were calculated from the 30 EPSCs at each individual Ca$^{2+}$ concentration. The variance was then plotted against the mean for each specific calcium condition using MATLAB software (MathWorks, USA). One additional data point, in which variance and mean are both theoretically at 0, was used for Ca$^{2+}$-free saline. Data from these five conditions were fit with a standard parabola (variance = $Q*\bar{I} - \bar{I}^2/N$), where Q is the quantal size, $\bar{I}$ is the mean evoked amplitude (x-axis), and N is the functional number of release sites. N, as a parameter of the standard parabola, was directly calculated for each cell by best parabolic fit.

## Statistical analysis

All data are presented as mean ± SEM. Data was compared using either a one-way ANOVA followed by Tukey's multiple comparison test, or using a Student's t-test (where specified). Data was analyzed using Graphpad Prism or Microsoft Excel software, with varying levels of significance assessed as $p<0.05$ (*), $p<0.01$ (**), $p<0.001$ (***), $p<0.0001$ (****), ns: not significant. M6 >GluRIIA$^{RNAi}$ results were compared to M6 > mCherry$^{RNAi}$ (*Figure 2—figure supplement 1*) and *G14/+*, in addition to $w^{1118}$; in no case did the control change the statistically significant result. See *Supplementary file 1* for further statistical details and values.

## Acknowledgements

We thank Brian McCabe (EPFL Brain Mind Institute, Lausanne, Switzerland) and Leslie Griffith (Brandeis University, Waltham, MA, USA) for sharing *Drosophila* stocks. We acknowledge the Developmental Studies Hybridoma Bank for antibodies used in this study, and the Bloomington Drosophila Stock Center for fly stocks. This work was supported by a grant from the National Institutes of Health

(NS091546) and research fellowships from the Alfred P Sloan, Ellison Medical, Whitehall, Mallinck-rodt, and Klingenstein-Simons Foundations to DKD.

## Additional information

### Funding

| Funder | Grant reference number | Author |
|---|---|---|
| National Institute of Neurological Disorders and Stroke | NS091546 | Dion K Dickman |
| Whitehall Foundation | | Dion K Dickman |
| Esther A. and Joseph Klingenstein Fund | Klingenstein-Simons Fellowship | Dion K Dickman |
| Simons Foundation | Klingenstein-Simons Fellowship | Dion K Dickman |
| Alfred P. Sloan Foundation | Research Fellowship in Neuroscience | Dion K Dickman |
| Ellison Medical Foundation | New Scholar in Aging | Dion K Dickman |
| Mallinckrodt | Junior Foundation Grant | Dion K Dickman |

The funders had no role in study design, data collection and interpretation, or the decision to submit the work for publication.

### Author contributions

Xiling Li, Conceptualization, Data curation, Formal analysis, Investigation, Methodology, Writing—original draft; Pragya Goel, Conceptualization, Methodology; Catherine Chen, Varun Angajala, Investigation, Methodology; Xun Chen, Data curation, Methodology; Dion K Dickman, Conceptualization, Supervision, Funding acquisition, Investigation, Methodology, Writing—original draft, Project administration, Writing—review and editing

### Author ORCIDs

Xiling Li (iD) http://orcid.org/0000-0002-2727-244X
Pragya Goel (iD) http://orcid.org/0000-0002-6348-4909
Dion K Dickman (iD) http://orcid.org/0000-0003-1884-284X

### Decision letter and Author response

Decision letter https://doi.org/10.7554/eLife.34338.016
Author response https://doi.org/10.7554/eLife.34338.017

## Additional files

### Supplementary files

• Supplementary file 1. Absolute values for all data and additional statistics. The figure and panel, genotype, and conditions used are noted (muscle segment, external calcium concentration, PhTx application). All electrophysiological recordings were performed in 0.4 mM external calcium unless specified otherwise. Average values for mEPSP, EPSP, quantal content, resting potential, input resistance, number of data samples (n), p values, and significance are shown. Average values for confocal imaging intensity levels are also indicated. Standard error values are noted in parentheses.
DOI: https://doi.org/10.7554/eLife.34338.013

• Transparent reporting form
DOI: https://doi.org/10.7554/eLife.34338.014

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
