## [Decision Letter]

Thank you for submitting your article "Synapse-specific and compartmentalized expression of presynaptic homeostatic potentiation" for consideration by *eLife*. Your article has been favorably evaluated by Eve Marder (Senior Editor) and three reviewers, one of whom, Ronald L Calabrese (Reviewer #1), is a member of our Board of Reviewing Editors. The following individual involved in review of your submission has agreed to reveal their identity: Richard Baines (Reviewer #3).

The reviewers have discussed the reviews with one another and the Reviewing Editor has drafted this decision to help you prepare a revised submission.

Summary:

Here the authors report elegant experiments that show that different terminals of the same motor neuron can undergo Presynaptic Homeostatic Plasticity (PHP) independently in response to knockdown of GluRIIA subunit on one muscle but not another also innervated by the same motor neuron. They present strong physiological evidence that the terminal of the motor neuron on the affected muscle undergo PHP do so by increasing the number of release sites and the readily releasable pool of vesicles specifically at boutons on the affected muscle. They also show that pCaMKII is down regulated opposite to Ib boutons on the affected muscle but not opposite Ib boutons on the other muscle. The experiments are very well controlled and supplementary data are germane, useful and well integrated. They argue that although the compartmentalized expression of PHP was not unexpected, given such early work as that by Hodge et al. 2006 on terminal specific pCaMKII expression, there was precedent to suspect inter-terminal crosstalk during homeostatic signaling and they have carefully ruled this out. The Discussion is informative and provocative suggesting how bouton specific postsynaptic retrograde signaling necessary to give the new results might occur.

The wider appeal of this work derives from how it fits with existing literature on synaptic plasticity and synaptic scaling in the vertebrate CNS providing an intriguing presynaptic alternative bouton specific mechanism of synaptic homeostatic scaling.

The writing is very clear and the figures nicely illustrative of the methods and the results.

Essential revisions:

There was a concern brought up vis a vis significance in that this study does not really extend current mechanistic understanding. If there is any place in this paper where this concern can be overcome, it would be to extend the result in Figure 5 (localized CaMKII de-phosphorylation as an instructive mechanism to induce PHP). This point is centered around the fact that the evidence is not air tight that CaMKII de-phosphorylation is actually driving a retrograde signal downstream of GluRIIA loss. In fact the authors appear to recognize this point and use qualified language, leaving open the possibility that CaMKII de-phosphorylation is just a sentinel for GluRIIA loss ("plausible mechanism" and "presumably related to PHP inductive signaling").

There are a few different ways this could be tested electrophysiologically. We would like you to consider whether one of these experiments is feasible within a short time-frame and whether you agree that it would strengthen the paper. If so, we would encourage you to do the experiment.

There are several possible experiments. Each would require a control electrophysiology data set and an experimental set. We think you have the expertise and resources to do one of these quickly.

a) M6 > UAS-CaMKII-ala vs. M6 > UAS-CaMKII-ala + > UAS-GluRIIA-RNAi. Record from both muscle 6 and muscle 7 as before. The expectation in muscle 6 would be that non-phosphorylatable CaMKII-ala instructively induces increased transmitter release in a way that is not additive with the addition of GluRIIA loss.

b) M6 > UAS-CaMKII-T287D vs. M6 > UAS-CaMKII-T287D + > UAS-GluRIIA RNAi. Record from both muscle 6 and muscle 7 as before. The expectation in muscle 6 would be that muscle expression of the constitutively phospho-mimetic version of CaMKII would block homeostatic potentiation of function. This is the experiment that the current manuscript says was done in Haghighi et al., 2003, but it was not really done because that paper does not check for baseline neurotransmission defects in T287D.

c) M6 > UAS-CamKII-ala vs. M6 > UAS-Tor vs. M6 > UAS-CaMKII-ala + > UAS-Tor. Record from both muscle 6 and 7. The expectation here is the same as in #1 above, non-additivity with muscle-driven UAS-Tor, which has been shown by multiple groups to induce presynaptic potentiation. The Dickman lab has the UAS-Tor line.

d) Show some evidence (any evidence) for the PSD/SSR model in the Discussion. With the M6 > UAS-GluRIIA-RNAi manipulation, it is a conundrum that GluRIIA is diminished at both Is and Ib boutons in muscle 6, but it is only at the Ib boutons where p-CaMKII is down. It is speculated that the PSD/SSR has something to do with this. At the Is boutons, Dlg is naturally diminished. So maybe the solution is to look at this problem in the context of globally diminished Dlg? We are less sure about what to expect here. The model is speculative.

There are several other points that should be addressed in the expert review but the Ca-imaging experiments suggested are not required.

*Reviewer #2:*

1) Driver Controls. The study relies on RNAi-mediated genetic knockdown of the GluRIIA subunit gene, using various muscle Gal4 manipulations to drive a UAS-GluRIIA^RNAi^ line. All of the electrophysiology and immunostaining and statistical analyses reported in the main figures utilize *w^1118^* as a control, as opposed to Gal4 (or RNAi line/+) alone. Driver controls would be better than *w^1118^* alone.

On the one hand, this does not pose a problem for the authors' main conclusions – one could view the Gal4/RNAi combination as a single, collective genetic perturbation that serves to reduce quantal size (or reduce CaMKII phosphorylation) versus *w^1118^*. On the other hand, since several of these reagents are being utilized and reported in the context of homeostatic plasticity for the first time (especially "M6-Gal4", which is comprised of multiple genetic manipulations), it would be better to report contributions of proper genetic controls in the main figures. It seems like the authors already have a better control already reported in the supplemental figure (M6-Gal4/mCherryRNAi). A G14/+ control would be also appropriate elsewhere.

2) CaMKII and mechanism. The authors follow up on prior work from their lab (Goel et al., 2017) and other labs (Newman et al., 2017; Hodge et al., 2006; Haghighi et al., 2003) to suggest that dephosphorylation of CaMKII in the muscle "mediates PHP induction". In the Discussion, much is offered about how this induction could potentially be working via CaMKII and the PSD (subsection “Compartmentalization of postsynaptic PHP signaling”, first paragraph).

It might be a nitpicky point, but the prior literature is not airtight on p-CaMKII's role in opposing retrograde signaling during PHP. The authors reference the T287D data from the Goodman lab (Haghighi et al., 2003). Those data were collected in a GluRIIA^SP16^ (or UAS-GluRIIAM/R) background, and baseline neurotransmission was not documented. Similarly, for studies utilizing the non-phospho UAS-CaMKIIala (T287A) reagent (Haghighi, Newman), it is not tested if the gains in neurotransmission are additive or non-additive in a GluRIIA null background. Muddying matters, there is other published work (e.g. Morimoto et al., Neuroscience 2010) that seems to suggest an opposite type of role for these gain- and loss-of-function alleles. That work invokes the opposite idea that CaMKII itself might be regulating GluR levels.

To be clear, the authors' demonstration that GluRIIA subunit loss promotes local CaMKII de-phosphorylation is solid data. It is a demonstration of a localized, visible response. That speaks to the main finding of the paper. However, it could still be the case that the CaMKII phenotype is reflective of muscle GluRIIA loss – and not actually an instructive part of homeostatic retrograde signaling itself. With their current data, the authors could qualify their discussion about CaMKII.

If the authors wished to extend the point about dephosphorylation of T287 CaMKII playing an instructive role in homeostatic potentiation, they could pursue other experiments. One idea would be to combine the Ala reagent with GluRIIA loss (or with Target of Rapamycin overexpression). For those genetic combinations, one might expect non-additivity in potentiation – as the Dickman lab recently demonstrated when combining GluRIIA loss and TOR overexpression (Goel et al., 2017).

3) Postsynaptic GluRIIA loss is muscle-specific, but not Ib-specific. The local versus global experiments in this paper provide compelling evidence that homeostatic potentiation signals do not "spread" (Introduction, end of first paragraph) from one presynaptic site Ib bouton to others presynaptic sites innervating the adjacent muscle. But the presynapse is not the only locus from which signals may spread. The GluRIIA loss perturbations are global (in a muscle-specific way). In theory, "homeostatic signals" (of any sort) could spread from postsynaptic Is sites to Ib sites within a given muscle, downstream of GluRIIA-containing receptor loss. Even though PHP is not expressed via the presynaptic Is boutons (Newman), and postsynaptic CaMKII phosphorylation is not dampened at Is loci, this idea could still be discussed.

4) Novelty of studying local presynaptic plasticity mechanisms. The Abstract and Introduction (first paragraph) state that it is uncertain whether plasticity mechanisms can be locally compartmentalized in a single presynaptic neuron's terminals. This is not quite correct.

The authors are correct that the question addressing PHP in this way has not been done before. But In the realm of synaptic plasticity (and NMJs in particular), there are many studies reaching back four decades or more that demonstrate target-specific presynaptic plasticity (see for example Frank, 1973; as well as Katz et al., 1993) at the lobster NMJ, terminals from a single neuron facilitate to varying degrees depending on the muscle contacted). Most relevant for the present study, the study cited for the H94 driver (Davis and Goodman, 1998) shows that forcing altered innervation between muscles 6 and 7 results in site-specific changes in neurotransmission. A broader plasticity context for the present work would be appropriate.

*Reviewer #3:*

My specific concerns are:

1) The GluRIIA RNAi would appear to be very effective. It would be a useful addition to quantitate this by QRT-PCR: perhaps using a pan-muscle GAL4 driver. Based on the IHC shown in Figure 2, one might predict KD to be >90% (which would be unusual, but not unheard of, for an RNAi).

2) Figure 6 reports the RRP and shows ejps elicited at 60Hz. The RRP is calculated, presumably, by dividing the cumulative ejp by mini size? One must presume, therefore, that minis were measured under voltage clamp? Mini data should be added to the figure to allow the reader to appreciate how the RRP was calculated. Alternatively, the precise manner in which RRP was calculated (in addition to relevant values) should be described.

3) With regard the theoretical calculation of N shown in Figure 7. Theory is one thing, but validation through experiment would be a welcome addition. Can the authors provide experimental validation – perhaps through use of GCaMP signalling (e.g. as per the Littleton lab) – to observe increased N?

---

## [Author Response]

Essential revisions:There was a concern brought up vis a vis significance in that this study does not really extend current mechanistic understanding. If there is any place in this paper where this concern can be overcome, it would be to extend the result in Figure 5 (localized CaMKII de-phosphorylation as an instructive mechanism to induce PHP). This point is centered around the fact that the evidence is not air tight that CaMKII de-phosphorylation is actually driving a retrograde signal downstream of GluRIIA loss. In fact the authors appear to recognize this point and use qualified language, leaving open the possibility that CaMKII de-phosphorylation is just a sentinel for GluRIIA loss ("plausible mechanism" and "presumably related to PHP inductive signaling").There are a few different ways this could be tested electrophysiologically. We would like you to consider whether one of these experiments is feasible within a short time-frame and whether you agree that it would strengthen the paper. If so, we would encourage you to do the experiment.There are several possible experiments. Each would require a control electrophysiology data set and an experimental set. We think you have the expertise and resources to do one of these quickly.a) M6 > UAS-CaMKII-ala vs. M6 > UAS-CaMKII-ala + > UAS-GluRIIA-RNAi. Record from both muscle 6 and muscle 7 as before. The expectation in muscle 6 would be that non-phosphorylatable CaMKII-ala instructively induces increased transmitter release in a way that is not additive with the addition of GluRIIA loss.b) M6 > UAS-CaMKII-T287D vs. M6 > UAS-CaMKII-T287D + > UAS-GluRIIA RNAi. Record from both muscle 6 and muscle 7 as before. The expectation in muscle 6 would be that muscle expression of the constitutively phospho-mimetic version of CaMKII would block homeostatic potentiation of function. This is the experiment that the current manuscript says was done in Haghighi et al., 2003, but it was not really done because that paper does not check for baseline neurotransmission defects in T287D.c) M6 > UAS-CamKII-ala vs. M6 > UAS-Tor vs. M6 > UAS-CaMKII-ala + > UAS-Tor. Record from both muscle 6 and 7. The expectation here is the same as in #1 above, non-additivity with muscle-driven UAS-Tor, which has been shown by multiple groups to induce presynaptic potentiation. The Dickman lab has the UAS-Tor line.d) Show some evidence (any evidence) for the PSD/SSR model in the Discussion. With the M6 > UAS-GluRIIA-RNAi manipulation, it is a conundrum that GluRIIA is diminished at both Is and Ib boutons in muscle 6, but it is only at the Ib boutons where p-CaMKII is down. It is speculated that the PSD/SSR has something to do with this. At the Is boutons, Dlg is naturally diminished. So maybe the solution is to look at this problem in the context of globally diminished Dlg? We are less sure about what to expect here. The model is speculative.

The reviewers request that we consider several possible experiments to provide mechanistic insight into the role of postsynaptic CaMKII and/or the subsynaptic reticulum opposite Ib bouton compartments in retrograde PHP signaling. In response, we have made attempts at all four proposed experiments. Although three of these experiments led to inconclusive results that require substantial additional work to properly interpret, one gave clear and conclusive insights regarding the requirement of CaMKII activity in driving PHP signaling following *GluRIIA* knock down. Details of this approach are discussed below.

M6>CaMKII-T287D +/- GluRIIA^RNAi^: As suggested by the reviewers, we overexpressed a transgene in the muscle, UAS-CaMKII-T287D, that constitutes a phospho-mimetic (constitutively active) version of CaMKII. Strong evidence for a required role of CaMKII in chronic PHP signaling was described in Haghighi et al., 2003, in which postsynaptic expression of this transgene was used to disrupt PHP expression in *GluRIIA* mutants. We expressed this transgene using the M6>Gal4 manipulation. While baseline function was not significantly impacted by M6>CaMKII-T287D, PHP failed to be expressed in M6>CaMKII-T287D+GluRIIA^RNAi^ (New Figure 8). This suggests that a reduction in postsynaptic CaMKII activity is necessary to induce retrograde PHP signaling, at least in the case of chronic genetic loss of the *GluRIIA* subunit.

These results are now detailed in an entirely new Figure 8, which we agree helps to extend mechanistic insight into the compartmentalized induction and expression of PHP. Given these new findings, we have altered the order of the figures in the revised manuscript to first describe the M6-Gal4 manipulation (Figures 1-2), then focus on compartmentalized presynaptic PHP expression mechanisms (Figures 3-6), and finally provide evidence for postsynaptic compartmentalized PHP induction via modulation of CaMKII activity (Figures 7-8). We thank the reviewers for pushing us to more fully consider and test the role of CaMKII in PHP signaling, which has substantially improved the manuscript.

There are several other points that should be addressed in the expert review but the Ca-imaging experiments suggested are not required.Reviewer #2:1) Driver Controls. The study relies on RNAi-mediated genetic knockdown of the GluRIIA subunit gene, using various muscle Gal4 manipulations to drive a UAS-GluRIIA^RNAi^ line. All of the electrophysiology and immunostaining and statistical analyses reported in the main figures utilize w1118 as a control, as opposed to Gal4 (or RNAi line/+) alone. Driver controls would be better than w1118 alone. On the one hand, this does not pose a problem for the authors' main conclusions – one could view the Gal4/RNAi combination as a single, collective genetic perturbation that serves to reduce quantal size (or reduce CaMKII phosphorylation) versus w1118. On the other hand, since several of these reagents are being utilized and reported in the context of homeostatic plasticity for the first time (especially "M6-Gal4", which is comprised of multiple genetic manipulations), it would be better to report contributions of proper genetic controls in the main figures. It seems like the authors already have a better control already reported in the supplemental figure (M6-Gal4/mCherryRNAi). A G14/+ control would be also appropriate elsewhere.

In Figure 2—figure supplement 1, we demonstrate that *w^1118^*, used throughout the manuscript as our “wild type” control, is not significantly different from an important genetic control for the M6-Gal4 manipulation: M6-Gal4>mCherry^RNAi^. This genotype controls for expression of an unrelated RNAi transgene (targeting mCherry), but otherwise includes the same genetic background and manipulations included in the M6>GluRIIA^RNAi^ used throughout the manuscript. In Figure 2—figure supplement 1, we show that muscle size, bouton number, active zone number, mEPSP amplitude, EPSP amplitude, and quantal content do not significantly differ in M6>mCherry^RNAi^ from *w^1118^*. We then use *w^1118^* as our control for the remainder of the manuscript. However, for additional thoroughness, we have also directly compared the results of M6>GluRIIA^RNAi^ shown in Figures 2, 3, and 4to M6>mCherry^RNAi^, which found the same statistically significant or insignificant differences as reported when compared to *w^1118^*. This is now discussed in the Materials and methods subsection “Statistical analysis”.

In addition, as suggested by the reviewer, we performed additional electrophysiological experiments using the driver only control (*G14-Gal4/+*). We found no significant difference in mEPSP amplitude, EPSP amplitude, or quantal content in *G14-Gal4/+* compared to *w^1118^*, but a significant difference was observed when compared to *GluRIIA* mutants and G14>GluRIIA^RNAi^, as expected. This is now discussed in the first paragraph of the subsection “Reduced expression of the glutamate receptor subunit *GluRIIA* specifically on muscle 6 at the *Drosophila* NMJ”and shown in Supplementary file 1. We thank the reviewer for these suggestions.

2) CaMKII and mechanism. The authors follow up on prior work from their lab (Goel et al., 2017) and other labs (Newman et al., 2017; Hodge et al., 2006; Haghighi et al., 2003) to suggest that dephosphorylation of CaMKII in the muscle "mediates PHP induction". In the Discussion, much is offered about how this induction could potentially be working via CaMKII and the PSD (subsection “Compartmentalization of postsynaptic PHP signaling”, first paragraph).It might be a nitpicky point, but the prior literature is not airtight on p-CaMKII's role in opposing retrograde signaling during PHP. The authors reference the T287D data from the Goodman lab (Haghighi et al., 2003). Those data were collected in a GluRIIA^SP16^ (or UAS-GluRIIAM/R) background, and baseline neurotransmission was not documented. Similarly, for studies utilizing the non-phospho UAS-CaMKIIala (T287A) reagent (Haghighi, Newman), it is not tested if the gains in neurotransmission are additive or non-additive in a GluRIIA null background. Muddying matters, there is other published work (e.g. Morimoto et al., Neuroscience 2010) that seems to suggest an opposite type of role for these gain- and loss-of-function alleles. That work invokes the opposite idea that CaMKII itself might be regulating GluR levels.To be clear, the authors' demonstration that GluRIIA subunit loss promotes local CaMKII de-phosphorylation is solid data. It is a demonstration of a localized, visible response. That speaks to the main finding of the paper. However, it could still be the case that the CaMKII phenotype is reflective of muscle GluRIIA loss – and not actually an instructive part of homeostatic retrograde signaling itself. With their current data, the authors could qualify their discussion about CaMKII.If the authors wished to extend the point about dephosphorylation of T287 CaMKII playing an instructive role in homeostatic potentiation, they could pursue other experiments. One idea would be to combine the Ala reagent with GluRIIA loss (or with Target of Rapamycin overexpression). For those genetic combinations, one might expect non-additivity in potentiation – as the Dickman lab recently demonstrated when combining GluRIIA loss and TOR overexpression (Goel et al., 2017).

We thank the reviewer for these suggestions and agree this would be of interest. We have responded in detail to these points in the “Essential revisions” section above.

3) Postsynaptic GluRIIA loss is muscle-specific, but not Ib-specific. The local versus global experiments in this paper provide compelling evidence that homeostatic potentiation signals do not "spread" (Introduction, end of first paragraph) from one presynaptic site Ib bouton to others presynaptic sites innervating the adjacent muscle. But the presynapse is not the only locus from which signals may spread. The GluRIIA loss perturbations are global (in a muscle-specific way). In theory, "homeostatic signals" (of any sort) could spread from postsynaptic Is sites to Ib sites within a given muscle, downstream of GluRIIA-containing receptor loss. Even though PHP is not expressed via the presynaptic Is boutons (Newman), and postsynaptic CaMKII phosphorylation is not dampened at Is loci, this idea could still be discussed.

We thank this reviewer for raising this valid point. We have now discussed this interesting idea in the revised Discussion (subsection “Compartmentalization of postsynaptic PHP signaling”, last paragraph).

4) Novelty of studying local presynaptic plasticity mechanisms. The Abstract and Introduction (first paragraph) state that it is uncertain whether plasticity mechanisms can be locally compartmentalized in a single presynaptic neuron's terminals. This is not quite correct.The authors are correct that the question addressing PHP in this way has not been done before. But In the realm of synaptic plasticity (and NMJs in particular), there are many studies reaching back four decades or more that demonstrate target-specific presynaptic plasticity (see for example Frank, 1973; as well as Katz et al., 1993) at the lobster NMJ, terminals from a single neuron facilitate to varying degrees depending on the muscle contacted). Most relevant for the present study, the study cited for the H94 driver (Davis and Goodman, 1998) shows that forcing altered innervation between muscles 6 and 7 results in site-specific changes in neurotransmission. A broader plasticity context for the present work would be appropriate.

We thank the reviewer for bringing these important studies to our attention. We agree they should be discussed as part of the framing of our manuscript. We have now revised the Introduction (first paragraph) to integrate these important studies and give credit for contributing insights into target-specific presynaptic plasticity mechanisms.

Reviewer #3:My specific concerns are:1) The GluRIIA RNAi would appear to be very effective. It would be a useful addition to quantitate this by QRT-PCR: perhaps using a pan-muscle GAL4 driver. Based on the IHC shown in Figure 2, one might predict KD to be >90% (which would be unusual, but not unheard of, for an RNAi).

We agree that the immunostaining results suggest a very effective *GluRIIA* knock down, and that quantification of this would be of interest. We utilized the newest generation of RNAi transgenes (TRiP lines) to target the *GluRIIA* subunit for knock down, and performed qPCR to measure levels of all 5 GluR subunits. This analysis revealed an 84% reduction of *GluRIIA* transcript levels in G14>GluRIIA^RNAi^, while transcripts encoding the other 4 GluR subunits were not significantly impacted. This data is now included in a new Figure 1 panel (Figure 1C) in the revised manuscript and discussed in the first paragraph of the subsection “Reduced expression of the glutamate receptor subunit *GluRIIA* specifically on muscle 6 at the *Drosophila* NMJ”. We thank the reviewer for this suggestion.

2) Figure 6 reports the RRP and shows ejps elicited at 60Hz. The RRP is calculated, presumably, by dividing the cumulative ejp by mini size? One must presume, therefore, that minis were measured under voltage clamp? Mini data should be added to the figure to allow the reader to appreciate how the RRP was calculated. Alternatively, the precise manner in which RRP was calculated (in addition to relevant values) should be described.

This concern was shared by reviewer 2. We have added the mEPSC values as a new panel in Figure 5 (Figure 5C), and thank the reviewer for this suggestion.

3) With regard the theoretical calculation of N shown in Figure 7. Theory is one thing, but validation through experiment would be a welcome addition. Can the authors provide experimental validation – perhaps through use of GCaMP signalling (e.g. as per the Littleton lab) – to observe increased N?

We agree this approach would be of interest. We have initiated calcium imaging studies in our lab using the SynaptoGCaMP6f reporter recently described in Newmann et al., 2017. Although the results are too preliminary to include in the present manuscript, we hope to include the outcomes of this approach in a future study. That being said, there is direct evidence for additional functional release sites recruited to participate in transmission at Ib boutons of *GluRIIA* mutants. This was reported in Newmann et al., 2017, in which they used calcium imaging to reveal what they referred to as “awakening” of release site number at Ib boutons during PHP. These imaging results are consistent with the mean/variance electrophysiological analysis reported in our manuscript (revised Figure 6).